# Neural signatures of vigilance decrements predict behavioural errors before they occur

Hamid Karimi-Rouzbahani[1,2,3]*, Alexandra Woolgar[1,2,3†], Anina N Rich[1,3†]

[1]Perception in Action Research Centre, Faculty of Human Sciences, Macquarie University, Sydney, Australia; [2]Medical Research Council Cognition and Brain Sciences Unit, University of Cambridge, Cambridge, United Kingdom; [3]Department of Cognitive Science, Faculty of Human Sciences, Macquarie University, Sydney, Australia

**Abstract** There are many monitoring environments, such as railway control, in which lapses of attention can have tragic consequences. Problematically, sustained monitoring for rare targets is difficult, with more misses and longer reaction times over time. What changes in the brain underpin these 'vigilance decrements'? We designed a multiple-object monitoring (MOM) paradigm to examine how the neural representation of information varied with target frequency and time performing the task. Behavioural performance decreased over time for the rare target (monitoring) condition, but not for a frequent target (active) condition. There was subtle evidence of this also in the neural decoding using Magnetoencephalography: for one time-window (of 80ms) coding of critical information declined more during monitoring versus active conditions. We developed new analyses that can predict behavioural errors from the neural data more than a second before they occurred. This facilitates pre-empting behavioural errors due to lapses in attention and provides new insight into the neural correlates of vigilance decrements.

*For correspondence:
hamid.karimi-rouzbahani@mrc-cbu.cam.ac.uk

†These authors contributed equally to this work

Competing interests: The authors declare that no competing interests exist.

## Introduction

When people monitor displays for rare targets, they are slower to respond and more likely to miss those targets relative to frequent target conditions (*Wolfe et al., 2005*; *Warm et al., 2008*; *Rich et al., 2008*; *Reason, 1990*; *Reason, 2000*). This effect is more pronounced as the time doing the task increases, which is often called a 'vigilance decrement'. Theoretical accounts of vigilance decrements fall into two main categories. 'Cognitive depletion' theories suggest performance drops as cognitive resources are 'used up' by the difficulty of sustaining attention under vigilance conditions (*Helton and Warm, 2008*; *Helton and Russell, 2011*; *Warm et al., 2008*). In contrast, 'mind wandering' theories suggest that the boredom of the task tends to result in insufficient involvement of cognitive resources, which in turn leads to performance decrements (*Manly et al., 1999*; *Smallwood and Schooler, 2006*; *Young and Stanton, 2002*). Either way, there are many real-life situations where such a decrease in performance over time can lead to tragic consequences, such as the Paddington railway disaster (UK, 1999), in which a slow response time to a stop signal resulted in a train moving another 600 m past the signal into the path of an oncoming train. With the move towards automated and semi-automated systems in many high-risk domains (e.g., power-generation and trains), humans now commonly need to monitor systems for infrequent computer failures or errors. These modern environments challenge our attentional systems and make it urgent to understand the way in which monitoring conditions change the way important information about the task is encoded in the human brain.

To date, most vigilance and rare target studies have used simple displays with static stimuli. Traditional vigilance tasks, inspired by radar operators in WWII, require participants to respond to infrequent visual events on otherwise blank screens, and show more targets are missed as time on task increases (*Mackworth, 1948*). More recent vigilance tasks have participants detect infrequent target stimuli among non-targets, and typically show an increase in misses as time on task increases. In *Temple et al., 2000*, for example, with only 20% targets, after 10 min target detection rates declined from 97% to 93% for high contrast (easy) and from 95% to 83% for low (hard) contrast targets. Other approaches have been to test for vigilance effects using frequent responses to non-targets, which have the advantage of more data points for analysis. The Sustained Attention to Response Task (SART), for example, requires participants to respond to each non-target item in a rapid stream of stimuli and occasionally withhold a response to a target item (*Beck et al., 1956*; *Rosenberg et al., 2013*). These approaches usually show effects on reaction times (RTs), which increase and become more variable with time on task (*Rosenberg et al., 2013*; *Möckel et al., 2015*; *Singleton, 1953*), although others have found RTs decrease (*Rubinstein, 2020*). Faster RTs also occur for 'target absent' responses in rare target visual search (*Wolfe et al., 2005*; *Rich et al., 2008*). Overall, vigilance decrements in terms of poorer performance can be seen in both accuracy and in RTs, depending on the task.

Despite these efforts, modern environments (e.g., rail and air traffic control) have additional challenges not encapsulated by these measures. This includes multiple moving objects, potentially appearing at different times, and moving simultaneously in different directions. When an object moves in the space, its neural representation has to be continuously updated so we can perceive the object as having the same identity. Tracking moving objects also requires considerable neural computation: in addition to spatial remapping, for example, we need to predict direction, speed, and the distance of the object to a particular destination. These features cannot be studied using static stimuli; they require objects that shift across space over time. In addition, operators have complex displays requiring selection of some items while ignoring others. We therefore need new approaches to study vigilance decrements in situations that more closely resemble the real-life environments in which humans are now operating. Developing these methods will provide a new perspective on fundamental questions of how the brain implements sustained attention in moving displays, and the way in which monitoring changes the encoding of information compared with active task involvement. These new methods may also provide avenues to optimise performance in high-risk monitoring environments.

The brain regions involved in maintaining attention over time has been studied using functional magnetic resonance imaging (fMRI), which measures changes in cerebral blood flow (*Adler et al., 2001*; *Benedict et al., 2002*; *Coull et al., 1996*; *Gilbert et al., 2006*; *Johannsen et al., 1997*; *Ortuño et al., 2002*; *Périn et al., 2010*; *Schnell et al., 2007*; *Sturm et al., 1999*; *Tana et al., 2010*; *Thakral and Slotnick, 2009*; *Wingen et al., 2008*). These studies compared brain activation in task vs. resting baseline or sensorimotor control (which involved no action) conditions and used univariate analyses to identify regions with higher activation under task conditions. This has the limitation that there are many features that differ between the contrasted (subtracted) conditions, not just the matter of sustained attention. Specifically, this comparison cannot distinguish whether the activation during sustained attention is caused by the differences in the task, stimuli, responses, or a combination of these factors. As it is challenging to get sufficient data from monitoring (vigilance) tasks in the scanner, many previous studies used tasks with relatively frequent targets, in which vigilance decrements usually do not occur. However, despite these challenges, *Langner and Eickhoff, 2013* reviewed vigilance neuroimaging studies and identified a network of right-lateralised brain regions including dorsomedial, mid- and ventrolateral prefrontal cortex, anterior insula, parietal and a few subcortical areas that they argue form the core network subserving vigilant attention in humans. The areas identified by *Langner and Eickhoff, 2013* show considerable overlap with a network previously identified as being recruited by many cognitively challenging tasks, the 'multiple demand' (MD) regions, which include the right inferior frontal gyrus, anterior insula, and intra-parietal sulcus (*Duncan and Owen, 2000*; *Duncan, 2010*; *Fedorenko et al., 2013*; *Woolgar et al., 2011*; *Woolgar et al., 2015a*; *Woolgar et al., 2015b*).

Other fMRI studies of vigilance have focused on the default mode network, composed of discrete areas in the lateral and medial parietal, medial prefrontal, and medial and lateral temporal cortices such as posterior cingulate cortex (PCC) and ventral anterior cingulate cortex (vACC), which is

thought to be active during 'resting state' and less active during tasks (*Greicius et al., 2003*; *Greicius et al., 2009*; *Raichle, 2015*). *Eichele et al., 2008* suggested that lapses in attention can be predicted by decrease of deactivation of this default mode network. In contrast, *Weissman et al., 2006* identified deactivation in the anterior cingulate and right prefrontal regions in pre-stimulus time windows when targets were missed. *Ekman et al., 2012* also observed decreased connectivity between sensory visual areas and frontal brain areas on the pre-stimulus time span of incorrect trials in colour/motion judgement tasks. More recently, *Sadaghiani et al., 2015* showed that the functional connectivity between sensory and 'vigilance-related' (cingulo-opercular) brain areas decreased prior to behavioural misses in an auditory task while between the same sensory area and the default-mode network the connectivity increased. These findings suggest that modulation of interactions between sensory and vigilance-related brain areas might be related to behavioural misses in monitoring tasks.

Detecting changes in brain activation that correlate with lapses of attention can be particularly challenging with fMRI, given that it has poor temporal resolution. Electroencephalography (EEG), which records electrical activity at the scalp, has much better temporal resolution, and has been the other major approach for examining changes in brain activity during sustained attention tasks. Frequency band analyses have shown that low-frequency alpha (8–10.9 Hz) oscillations predict task workload and performance during monitoring of simulated air traffic (static) displays with rare targets, while frontal theta band (4–7.9 Hz) activity predicts task workload only in later stages of the experiment (*Kamzanova et al., 2014*). Other studies find that increases in occipital alpha oscillations can predict upcoming error responses (*Mazaheri et al., 2009*) and misses (*O'Connell et al., 2009*) in go/no-go visual tasks with target frequencies of 11% and 9%, respectively. These changes in signal power that correlate with the task workload or behavioural outcome of trials are useful, but provide relatively coarse-level information about what changes in the brain during vigilance decrements.

Understanding the neural basis of decreases in performance over time under vigilance conditions is not just theoretically important, it also has potential real-world applications. In particular, if we could identify a reliable neural signature of attentional lapses, then we could potentially intervene prior to any overt error. For example, with the development of autonomous vehicles, being able to detect when a driver is not engaged, combined with information about a potential threat, could allow emergency braking procedures to be initiated. Previous studies have used physiological measures such as pupil size (*Yoss et al., 1970*), body temperature (*Molina et al., 2019*), skin conductance, and blood pressure (*Lohani et al., 2019*) to indicate the level of human arousal or alertness, but these lack the fine-grained information necessary to distinguish transient dips from problematic levels of inattention in which task-related information is lost. In particular, we lack detail on how information processing changes in the brain during vigilance decrements. This knowledge is crucial to develop a greater understanding of how humans sustain vigilance.

In this study, we developed a new task, multiple-object monitoring (MOM), which includes key features of real-life situations confronting human operators in high-risk environments. These features include moving objects, varying levels of target frequency, and a requirement to detect and avoid collisions. A key feature of our MOM task is that it allows measurement of the specific decrements in performance during vigilance (sustaining attention in a situation where only infrequent responses are needed) separate from more general decreases in performance simply due to doing a task for an extended period. Surprisingly, this is not typically the case in vigilance tasks. We recorded neural data using the highly sensitive method of magnetoencephalography (*Baillet, 2017*) and used multivariate pattern analyses (MVPA) to determine how behavioural vigilance decrements correlate with changes in the neural representation of information. We used these new approaches to better understand the way in which changes between active and monitoring tasks affect neural representation, including functional connectivity. We then examined the potential for using these neural measures to predict forthcoming behavioural misses based on brain activity.

## Results

Participants completed the MOM task during which they monitored several dots moving on visible trajectories towards a centrally presented fixed object (*Figure 1A*). The trajectories spanned from corners of the screen towards the central object and deflected at 90° before contacting the central object. The participants' task was to keep fixation on the central object and press the button to

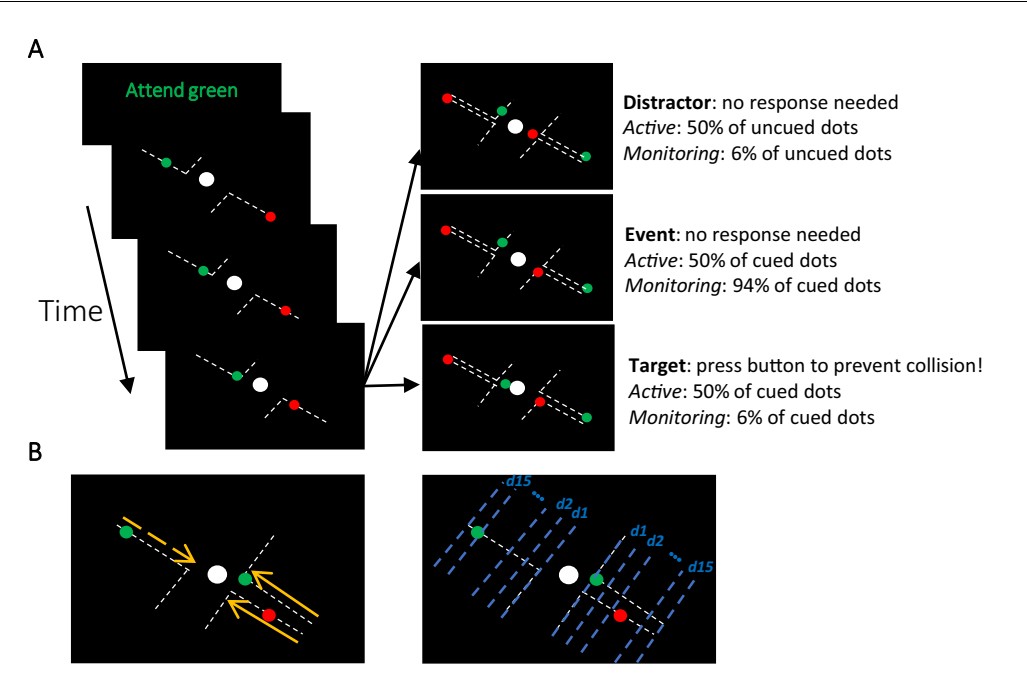

**Figure 1.** The multiple-object monitoring (MOM) task and types of information decoded. (**A**) At the start of a block, the relevant colour is cued (here, green; distractors in red). Over the on-task period (~30 min per task condition), multiple dots entered from either direction, each moving along a visible individual trajectory towards the middle object. Only attended dots that failed to deflect along the trajectories at the deflection point required a response (Target: bottom right display). Participants did not need to press the button for the unattended dot (Distractor: top right display) or the dots that kept moving on the trajectories (Event: middle right panel). Each dot took ~1226 ms from appearance to deflection. (**B**) Direction of approach information (left display: left vs. right as indicated by dashed and solid lines, respectively) and distance to object information (right display). Note the blue dashed lines and orange arrows were not present in the actual display. d1, d2, etc. denote the 'distance units' used to train the classifier for the key distance to object information. A demo of the task can be found here [https://osf.io/5aw8v/].

deflect the moving dot if it violated its trajectory and continued towards the central object after reaching the deflection point. They were tasked to do so before the object 'collided' with the central object. In each block, only dots of one colour (either green or red; called Attended vs. Unattended) was relevant and should be responded to by the participants (~110 s). Either 50% or 6% of the attended dots (cued colour) were targets (i.e., violated their trajectory requiring a response; see Materials and methods) generating *Active* and *Monitoring* conditions, respectively.

## Behavioural data: The MOM task evokes a reliable vigilance decrement

In the first block of trials (i.e., the first 110 s, excluding the two practice blocks), participants missed 29% of targets in the Active condition and 40% of targets in the Monitoring condition. However, note the number of targets in any single block is necessarily very low for the Monitoring (for a single block, there are 16 targets for Active but only two targets for Monitoring). The pattern becomes more robust over blocks, and *Figure 2A* shows the miss rates changed over time in different directions for the Active vs. Monitoring conditions. For Active blocks, miss rates decreased over the first five blocks and then plateaued at ~17%. For Monitoring, however, miss rates increased throughout the experiment: by the final block, these miss rates were up to 76% (but again, the low number of targets in Monitoring mean that we should use caution in interpreting the results of any single block alone). There was evidence that miss rates were higher in the Monitoring than Active conditions from the fourth block onwards (BF >3; *Figure 2A*). Participants' RTs on *correct* trials also showed evidence of specific vigilance decrements, increasing over time under Monitoring but decreasing under Active task conditions (*Figure 2B*). There was evidence that RTs were slower for Monitoring

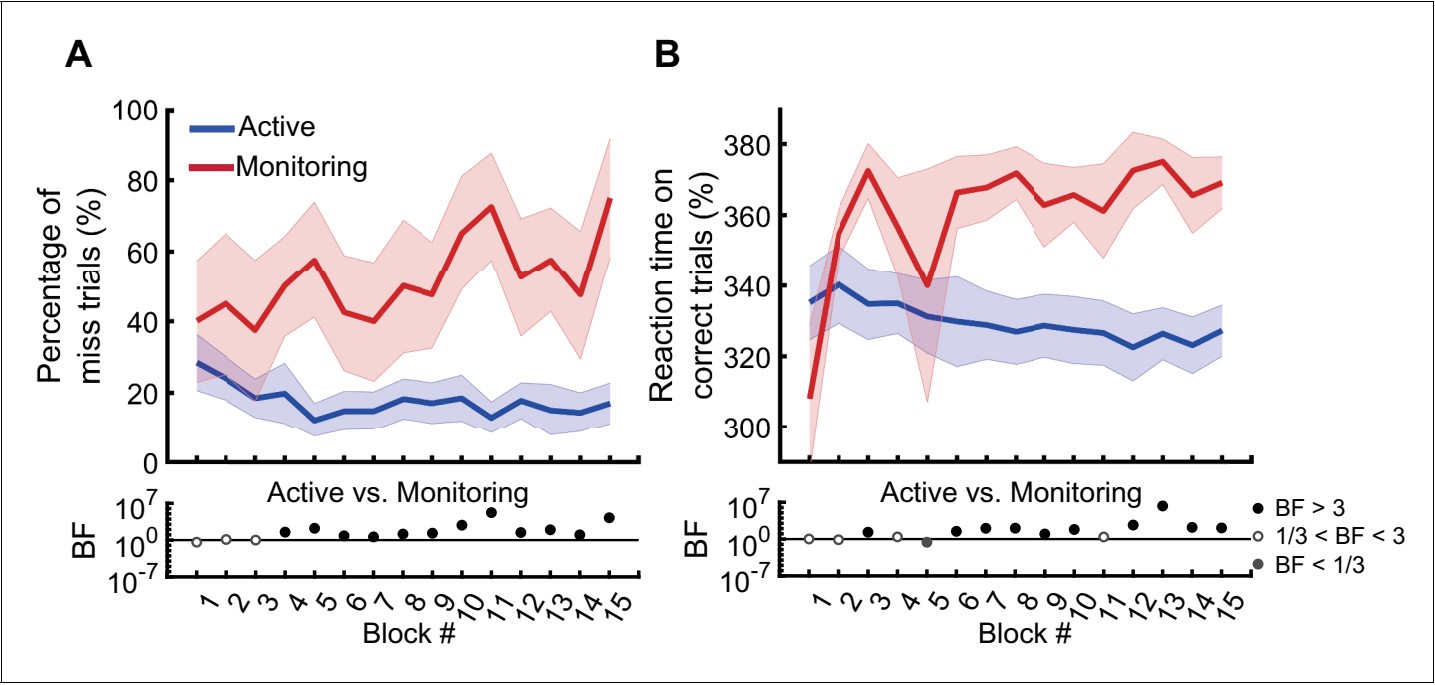

**Figure 2.** Behavioural performance on the MOM task. The percentage of miss trials (A), and correct reaction times (B), as a function of block. Thick lines show the average across participants (shading 95% confidence intervals) for Active (blue) and Monitoring (red) conditions. Each block lasted for 110 s and had either 16 (Active) or 2 (Monitoring) targets out of 32 cued-colour and 32 non-cued colour dots. Bayes factors (BF) are shown in the bottom section of each graph: Filled circles show moderate/strong evidence for either hypothesis and empty circles indicate insufficient evidence when evaluating the contrast between Active and Monitoring conditions.

compared with Active from the sixth block onwards (BF >3, except for Block #11). The characteristic pattern of increasing miss rates and slower RTs over time in the Monitoring relative to the Active condition validates the MOM task as effectively evoking vigilance decrements.

## Neural data: Decoding different aspects of task-related information

We used multivariate pattern analysis (i.e., decoding) to extract two types of information from MEG data about each dot's movement on the screen: information about the *direction of approach* (whether the dot was approaching the central object from left or right side of the screen) and *distance to object* (how far was the dot relative to the central object; *Figure 1B*; see Materials and methods).

With so much going on in the display at one time, we first needed to verify that we can successfully decode the major aspects of the moving stimuli, relative to chance. The full data figures and details are presented in Supplementary materials: We were able to decode both *direction of approach* and *distance to object* from MEG signals (see *Figure 3—figure supplement 1*). Thus, we can turn to our main question about how these representations were affected by the Target Frequency, Attention, and Time on Task.

## The neural correlates of the vigilance decrement

As the behavioural results showed (*Figure 2*), the difference between Active and Monitoring conditions increased over time, showing the greatest difference during the final blocks of the experiment. To explore the neural correlates of these vigilance decrements, we evaluated information representation in the brain during the first five and last five blocks of each task (called early and late blocks, respectively) and the interactions between the Target Frequency, Attention, and the Time on Task using a three-way Bayes factor ANOVA.

## Effects of target frequency on *direction of approach* information

*Direction of approach* information is a very clear visual signal ('from the left' vs. 'from the right') and therefore is unlikely to be strongly modulated by other factors, except perhaps whether the dot was in the cued colour (Attended) or the distractor colour (could be ignored: Unattended). There was moderate or strong evidence for a main effect of Attention (*Figure 3A*; BF >3, Bayes factor ANOVA, cyan dots) starting from 265 ms and lasting until dots faded. This is consistent with maintenance of information about the attended dots and attenuation of the information about unattended dots (*Figure 3—figure supplement 1A*). The large difference in coding attributable to attention remained for as long as the dots were visible.

In contrast, there was no sustained main effect of Target Frequency on the same *direction of approach* coding. For the majority of the epoch there was moderate or strong evidence for the null hypothesis (BF <1/3; Bayes factor ANOVA, *Figure 3A*, purple dots). The sporadic time point with a main effect of Target Frequency, observed before the deflection (BF >3), likely reflects noise in the data as there is no clustering. Recall that we only focus on time points prior to deflection, as after this point there are visual differences between Active and Monitoring, with more dots deflecting in the Monitoring condition.

There was also no sustained main effect of the Time on Task on information about the *direction of approach* (BF <3; Bayes factor ANOVA, green dots; *Figure 3A*). There were no sustained two-way or three-way interactions between Attention, Target Frequency, and Time on Task (BF <3; Bayes factor ANOVA). Note that the number of trials used in the training and testing of the classifiers were equalised across the eight conditions and equalled the minimum available number of trials across those conditions shown in *Figure 3*. Therefore, the observed effects cannot be attributed to a difference in the number of trials across conditions.

## Effects of target frequency on critical *distance to object* information

The same analysis for the representation of the task-relevant *distance to object* information showed strong evidence for a main effect of Attention (BF > 10; Bayes factor ANOVA) at all 15 distances, no effect of Time on Task (BF < 0.3; Bayes factor ANOVA) at any of the distances, and an interaction between Time on Task and Target Frequency at one of the distances (BF = 6.7, *Figure 3B*). The interaction between Target Frequency and Time on Task at distance 13 (time-window: 160 to 240 ms after stimulus onset, BF = 6.7) reflected opposite effects of time on task in the Active and Monitoring conditions. In Active blocks, there was moderate evidence that coding was stronger in late blocks than in early blocks (BF = 3.1), whereas in the Monitoring condition, decoding declined with time and was weaker in late than in easy blocks (BF = 4.3). However, as there was only moderate evidence for this interaction at one of the time-windows, we do not overinterpret it. Decoding of attended information tended to be lower in late compared to early Monitoring blocks (Figure 3B lower panel red dotted line) in several time-windows across the trial, which may echo the behavioural pattern of performance (Figure 2). As there was moderate evidence for no interaction between Attention and Target Frequency (BF < 0.3, 2-way Bayes factor ANOVA) except for distance 6 (BF = 3.3; no consistent pattern (insufficient evidence for pairwise comparisons: BFs 2.4-2.8)), no interaction between Attention and Time on Task (BF < 0.3, 2-way Bayes factor ANOVA) or simultaneously between the three factors (BF < 0.3, 3-way Bayes factor ANOVA), we do not show those statistical results in the figure.

Although eye-movements should not drive the classifiers due to our design, it is still important to verify that the results replicate when standard artefact removal is applied. We can also use eye-movement data as an additional measure, examining blinks, saccades and fixations for effects of our attention and vigilance manipulations.

First, to make sure that our neural decoding results replicate after eye-related artefact removal, we repeated our analyses on the data after eye-artefact removal, which provided analogous results to the original analysis (see the decoding results with and without artefact removal in *Figure 3—figure supplement 2*). Specifically, for our crucial *distance to object* data, the main effect of Attention remained after eye-artefact removal, replicating our initial pattern of results. Moderate evidence (BF = 4.2) for an interaction between Target Frequency and Time on Task was also found, but now at distance 6 instead of distance 13. This interaction again reflected a larger effect of Time on Task

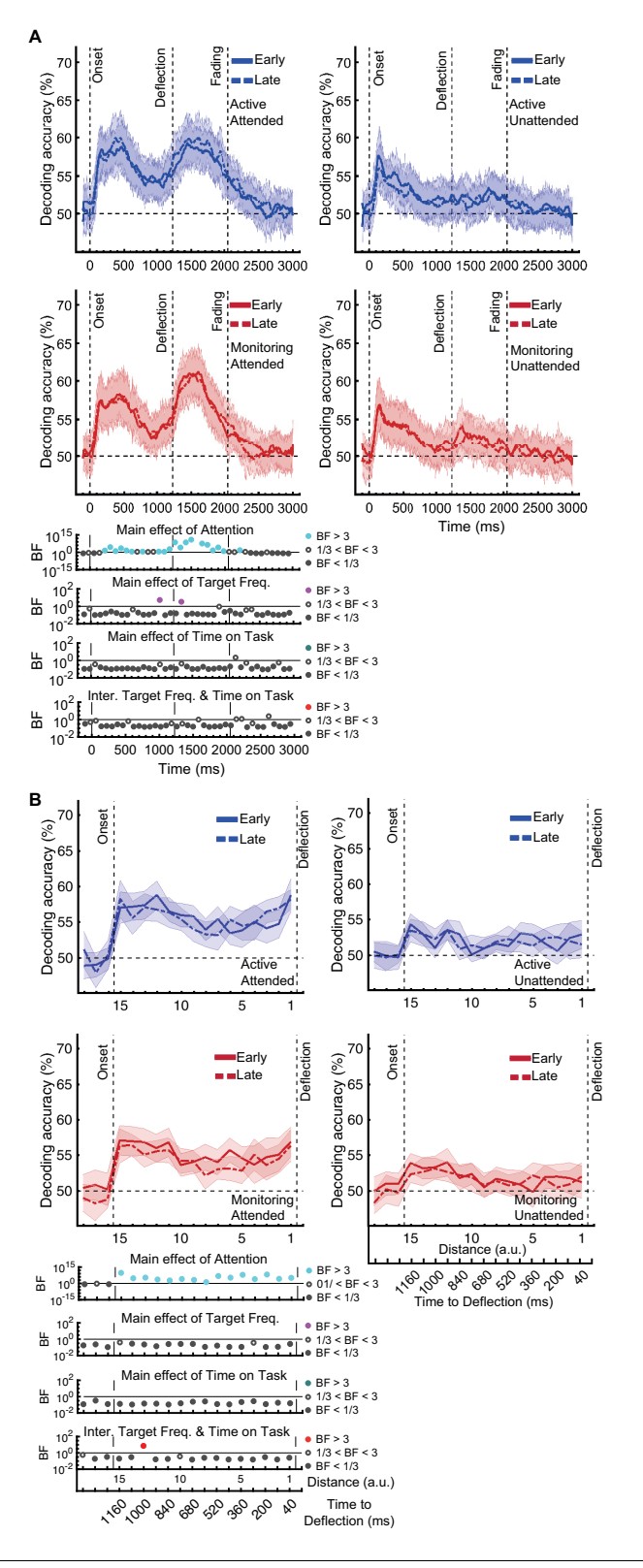

**Figure 3.** Impact of different conditions and their interactions on information on correct trials (all trials except those in which a target was missed or there was a false alarm). (**A**) Decoding of direction of approach information (less task-relevant) and (**B**) decoding of distance to object information (most task-relevant). Left two columns: Attended dots; Right two columns: Unattended ('distractor') dots. Thick lines show the average across participants

*Figure 3 continued on next page*

*Figure 3 continued*

(shading 95% confidence intervals). Horizontal dashed line refers to theoretical chance-level decoding (50%). Vertical dashed lines indicate critical times in the trial. Bottom panels: Bayesian evidence for main effects and interactions, Bayes factors (BF): Filled circles show moderate/strong evidence for either hypothesis and empty circles indicate insufficient evidence. Main effects and interactions of conditions calculated using BF ANOVA analysis. Cyan, purple, green, and red dots indicate the main effects of Attention, Target frequency, Time on Task, and the interaction between Target frequency and Time on Task, respectively. The results of BF analysis (i.e., the main effects of the three conditions and their interactions) are from the same three-way ANOVA analysis and are therefore identical for attended and unattended panels. Early = data from the first five blocks (~10 min). Late = data from the last five blocks (~10 min). Note the different scales of the BF panels, and the down-sampling, for clearer illustration.

The online version of this article includes the following figure supplement(s) for figure 3:

**Figure supplement 1.** Impact of different conditions in the direction of approach and distance to object information coding and their Bayesian evidence for difference from chance.

**Figure supplement 2.** Impact of different conditions and their interactions on information processing on correct trials (all trials except those in which a target was missed or there was a false alarm) without and with standard eye-artefact removal.

**Figure supplement 3.** The statistical relationship between the proportion of eye-related measures and Target Frequency, Attention, and Time on the task factors.

in Monitoring compared to Active blocks (Monitoring: weaker coding in late relative to early blocks (BF = 3.1); Active: insufficient evidence for change in coding from early to late (BF = 2.0)).

Second, we conducted a post hoc analysis to explore whether eye movement data showed the same patterns of vigilance decrements and therefore could explain our decoding results. We extracted the proportion of eye blinks, saccades, and fixations per trial as well as the duration of those fixations from the eye-tracking data for *correct* trials (−100 to 1400 ms aligned to the stimulus onset time), and statistically compared them across our critical conditions (*Figure 3—figure supplement 3*). We saw strong evidence (BF = $4.8e^8$) for a difference in the number of eye blinks between attention conditions: There were more eye blinks for the Unattended (distractor) than Attended (potentially targets) colour dots. We also observed moderate evidence (BF = 3.4) for difference between the number of fixations, with more fixations in Unattended vs. Attended conditions. These suggest that there are systematic differences in the number of eye blinks and fixations due to our attentional manipulation, consistent with previous observations showing that the frequency of eye blinks can be affected by the level of attentional recruitment (*Nakano et al., 2013*). However, there was either insufficient evidence (0.3 < BF <3) or moderate or strong evidence for no differences (0.1 < BF <0.3 and BF <0.3, respectively) between the number of eye blinks and saccades across our Active, Monitoring, Early, and Late blocks, where we observed our 'vigilance decrement' effects in decoding. Therefore, this suggests that the main vigilance decrement effects in decoding, which were evident as an interaction between Target frequency (Active vs. Monitoring) and Time on the task (Early vs. Late; *Figure 3*), are not primarily driven by eye movements. However, artefact removal algorithms are not perfect, making it is impossible to fully rule out all potentially meaningful eye-related artefacts from the MEG data (e.g. the difference in the number of eye blinks between attended and unattended conditions). Thus, although the results are similar with and without standard eye-artefact removal, it is impossible to fully rule out all potential eye movement effects.

Together, these results suggest that while vigilance conditions had little or no impact on coding of the *direction of approach*, they did impact the critically task-relevant information about the *distance* of the dot from the object, albeit only for one 80ms time-window. In this time-window, coding declined as time on task increased specifically when the target events happened infrequently, forming a possible neural correlate for our behavioural vigilance decrements.

## Is informational brain connectivity modulated by Attention, Target Frequency, and Time on Task?

Using graph-theory-based univariate connectivity analysis, it has been shown that the connectivity between relevant sensory areas and 'vigilance-related' cognitive areas changes prior to lapses in attention (behavioural errors; *Ekman et al., 2012*; *Sadaghiani et al., 2015*). Therefore, we asked whether vigilance decrements across the time course of our task corresponded to changes in

multivariate informational connectivity, which evaluates the similarity of information encoding, between frontal attentional networks and sensory visual areas. In line with attentional effects on sensory perception, we predicted that connectivity between the frontal attentional and sensory networks should be lower when not attending (vs. attending; *Goddard et al., 2019*). Behavioural errors were also previously predicted by reduced connection between sensory and 'vigilance-related' frontal brain areas (*Ekman et al., 2012*; *Sadaghiani et al., 2015*). Therefore, we predicted a decline in connectivity when targets were lower in frequency, and with increased time on task, as these led to increased errors in behaviour, specifically under vigilance conditions in our task (i.e., late blocks in Monitoring vs. late blocks in Active; *Figure 2*). We used a simplified version of our method of RSA-based informational connectivity to evaluate the (Spearman's rank) correlation between *distance* information RDMs across the peri-frontal and peri-occipital electrodes (*Goddard et al., 2016*; *Figure 4A*).

Results showed strong evidence (Bayes factor ANOVA, BF = 6.5e$^3$) for higher informational connectivity for trials with Attended compared to Unattended dots, and moderate evidence for no effect of Target Frequency (Bayes factor ANOVA, BF = 0.11; *Figure 4B*). There was insufficient evidence to determine whether there was a main effect of Time on Task (Bayes factor ANOVA, BF = 0.72). There was evidence in the direction of the null for the two-way interactions between the

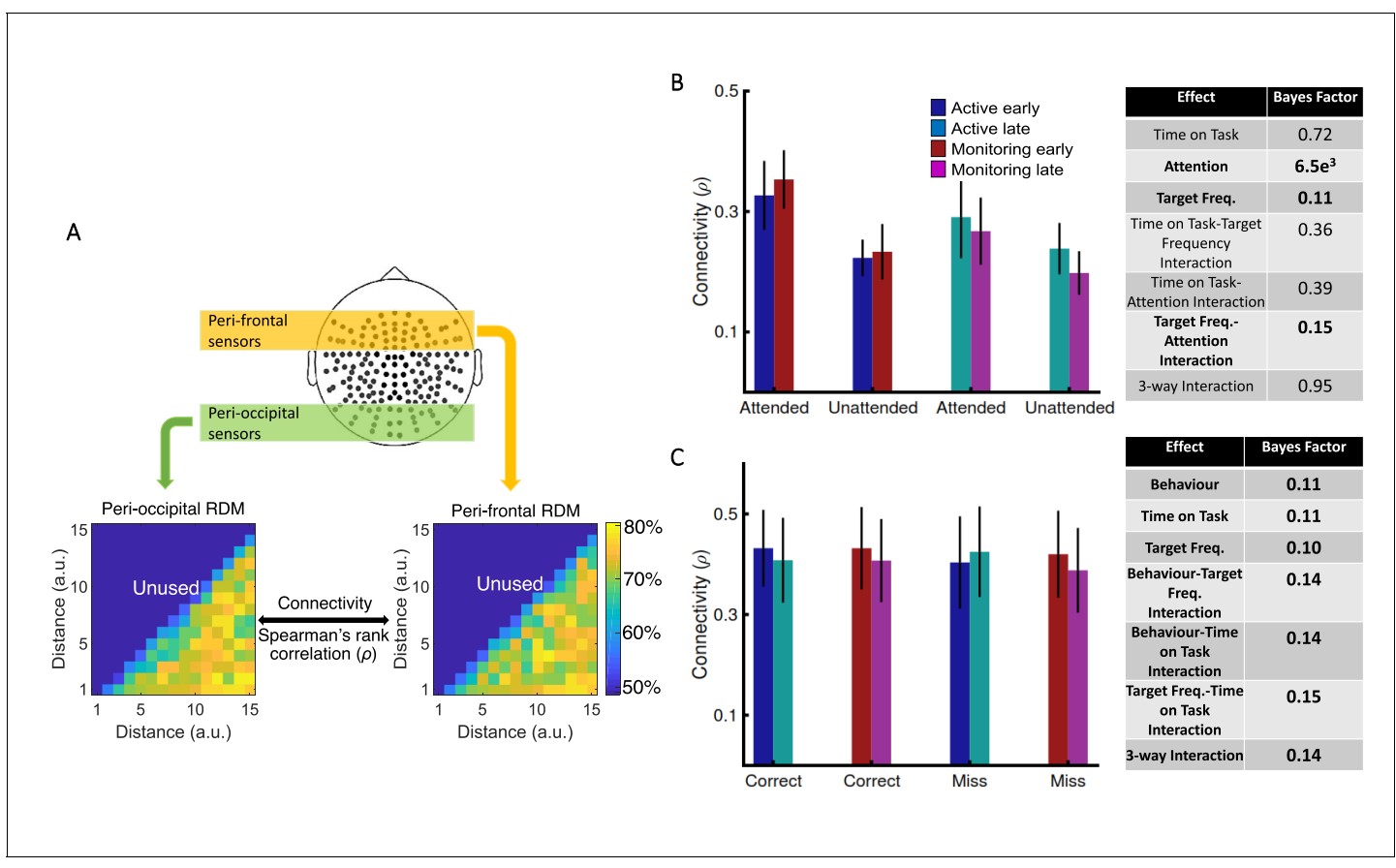

**Figure 4.** Relationship between informational connectivity and Attention, Target Frequency, Time on Task, and the behavioural outcome of the trial (i. e., correct vs. miss). (**A**) Calculation of connectivity using Spearman's rank correlation between RDMs obtained from the peri-frontal and peri-occipital sensors as indicated by coloured boxes, respectively. RDMs include decoding accuracies obtained from testing the 105 classifiers trained to discriminate different distance to object categories. (**B**) Connectivity values for the eight different conditions of the task and the results of three-way Bayes factor ANOVA with factors Time on Task (Early, Late), Attention (Attended, Unattended), and Target Frequency (Active, Monitoring), using only correct trials. (**C**) Connectivity values for the Active and Monitoring, Early and Late blocks of each task for correct and miss trials (attended condition only), and the result of Bayes factor ANOVA with factors Target Frequency (Active, Monitoring), Time on Task (Early, Late), and behavioural outcome (correct, miss) as inputs. Number of trials are equalised across conditions in **B** and **C** separately. Bars show the average across participants (error bars 95% confidence intervals). Bold fonts indicate moderate or strong evidence for either the effect or the null hypothesis.

factors (Bayes factor ANOVA, two-way Time on Task-Target Frequency: BF = 0.36; Time on Task-Attention: BF = 0.39; Target Frequency-Attention: BF = 0.15) and insufficient evidence regarding their three-way interaction (BF = 0.95). These results suggest that [–deleted text–] trials in which the dots are in the distractor (Unattended) colour, in which the attentional load is low, result in less informational connectivity between occipital and frontal brain areas compared to [–deleted text–] Attended trials. This is consistent with a previous study (*Alnæs et al., 2015*), which suggested that large-scale functional brain connectivity depends on the attentional load, and might underpin or accompany the decrease in information decoding across the brain in the unattended condition., which suggested that large-scale functional brain connectivity depends on the attentional load, and might underpin or accompany the decrease in information decoding across the brain in these conditions.

We also compared the connectivity for the *correct* vs. *miss* trials (*Figure 4C*). This analysis was performed only for Attended condition as there are no *miss* trials for Unattended condition, by definition. There was moderate evidence for no difference in connectivity on *miss* compared to *correct* trials (Bayes factor ANOVA, BF = 0.11). In addition, there was moderate evidence for no effect of Time on Task and Target Frequency (BF = 0.11 and BF = 0.10, respectively), as well as for two-way and three-way interactions between the three factors (Bayes factor ANOVA, Behaviour-Target Frequency: BF = 0.14; Behaviour-Time on Task: BF = 0.14; Target Frequency-Time on Task: BF = 0.15; their 3-way interaction BF = 0.14). Therefore, in contrast to an auditory monitoring task which showed decline in univariate graph-theoretic connectivity before behavioural errors (*Sadaghiani et al., 2015*), we observed no change in informational connectivity on error. Note that, the number of trials is equalized across the 8 conditions in each of our analyses separately.

## Is neural representation different on *miss* trials?

The results presented in *Figure 3*, which used only *correct* trials, showed changes due to target frequency to the representation of task-relevant information when the task was performed successfully. We next move on to our second question, which is whether these neural representations change when overt behaviour *is* affected, and therefore, whether we can use the neural activity as measured by MEG to predict behavioural errors before they occur. We used our method of error data analysis (*Woolgar et al., 2019*) to examine whether the patterns of information coding on *miss* trials differed from *correct* trials. For these analyses we used only attended dots, as unattended dots do not have behavioural responses, and we matched the total number of trials in our implementation of correct and miss classification.

First, we evaluated the representation of the less relevant information – the *direction of approach* measure (*Figure 5A*). The results for *correct* trials provided information dynamics very similar to the Attended condition in *Figure 3A*, except for higher overall decoding, which is explained by the inclusion of the data from the whole experiment (15 blocks) rather than just the five early and late blocks (note the number of trials is still matched to *miss* trials).

For the *direction of approach* information, there was moderate or strong evidence (i.e., BF >3) in both Active and Monitoring conditions after ~100 ms for above-chance decoding. However, when the classifiers were tested on *miss* trials, from onset to deflection, the pattern of information dynamics were different, even though we had matched the number of trials. Specifically, while the level of information was comparable to *correct* trials with spurious instances (but no sustained time windows) of difference (BF >3 as indicated by black dots) before 500 ms, decoding traces were much noisier for *miss* trials with more variation across trials and between nearby time points (*Figure 5A*). Note that after the deflection, the visual signal is different for *correct* and *miss* trials, so the difference between their decoding curves (BF >3) is not meaningful. These results suggest a noisier representation of *direction of approach* information for the missed dots compared to correctly deflected dots.

We then repeated the same procedure on the representation of the most task-relevant *distance to object* information on *correct* vs. *miss* trials (*Figure 5B*). On *correct* trials, the *distance* information for both Active and Monitoring conditions was above chance (*Figure 5B* left panels; BF > $10^4$). For *miss* trials, the corresponding *distance* information was still above chance (*Figure 5B* right panels; BF > $10^3$) but the direct comparison revealed that distance information dropped on miss trials compared to correct trials (*Figure 5B*; Black dots; BF >3 across all distances; Active and Monitoring results were averaged for *correct* and *miss* trials separately before Bayes analyses).

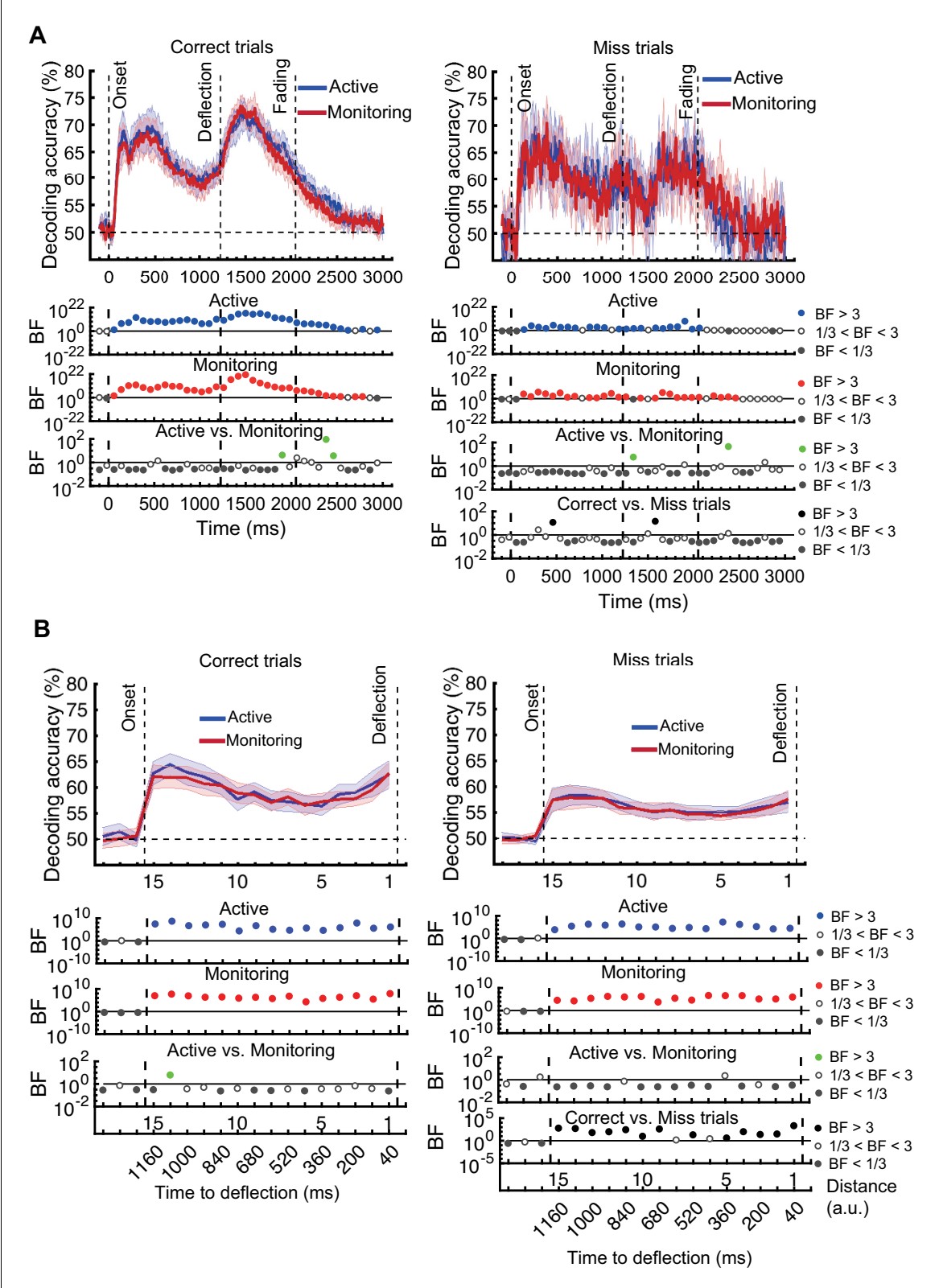

**Figure 5.** Decoding of information on correct vs miss trials. (**A**) Decoding of direction of approach information (less task-relevant). (**B**) Decoding of distance to object information (most task-relevant). The horizontal dashed lines refer to theoretical chance-level decoding. Left panels: Decoding using correct trials; Right panels: Decoding using miss trials. In both right and left panels, the classifiers were trained on correct trials and tested on (left out) correct and all miss trials, respectively. Thick lines show the average across participants (shading 95% confidence intervals). Vertical dashed lines

*Figure 5 continued on next page*

*Figure 5 continued*

indicate critical events in the trial. Bayes factors (BF) are shown in the bottom section of each graph: Filled circles show moderate/strong evidence for either hypothesis and empty circles indicate insufficient evidence. They show the results of BF analysis when evaluating the difference of the decoding values from chance for Active (blue) and Monitoring (red) conditions separately, the comparison of the two conditions (green), and the comparison of correct and miss trials (black). Note that for the comparison of correct and miss trials, Active and Monitoring conditions were averaged separately. Note the different scales of the BF panels, and down-sampling, for clearer illustration.

The online version of this article includes the following figure supplement(s) for figure 5:

**Figure supplement 1.** Distribution of decoding accuracies for every individual correct and miss trial in the Active and Monitoring conditions for all 21 subjects.

**Figure supplement 2.** Decoding of information on correct vs false alarm trials.

In principle, the average decoding levels could be composed of 'all or none' misses or graded drops in information, and it is possible that on some miss trials there is a good representation but the target is missed for other reasons (e.g., a response-level error). As neural data are noisy and multivariate decoding needs cross-validation across subsamples of the data, and because each trial, at each distance, can only be classified correctly or incorrectly by a two-way classifier, we tend not to compare the decoding accuracies in a trial-by-trial manner, but rather on average (*Grootswagers et al., 2017*). However, if we look at an individual data set and examine all the miss trials (averaged over the 15 distances and cross-validation runs) in our distance-to-object decoding, we can get some insights into the underlying distributions (*Figure 5—figure supplement 1*). Results showed that, for all participants, the distribution of classifier accuracies for both correct and miss trials followed approximate normal distributions. However, while the distribution of decoding accuracies for correct trials was centred around 60%, the decoding accuracies for individual miss trials were centred around 56%. We evaluated the difference in the distribution of classification accuracies between the two types of trials using Cohen's d. Cohen's d ranged from 0 to 2.5 across participants and conditions. 14 out of 21 subjects showed moderate (d > 0.5) to large (d > 0.8; *Cohen, 1969*) differences between the distribution of correct and miss trials in either Active or Monitoring condition or both. Therefore, although the miss trials vary somewhat in levels of information, only a minority of (< 24%) miss trials are as informative as the least informative correct trials. These results are consistent with the interpretation that there was less effective representation of the crucial information about the distance from the object preceding a behavioural miss.

Please note that the results presented so far were from *correct* and *miss* trials and we excluded early, late, and wrong-colour *false alarms* to be more specific about the error type. However, the false alarm results (collapsed across all three types of false alarms) were very similar (*Figure 5—figure supplement 2*) to those of the missed trials (*Figure 5*): noisy information about the *direction of approach* and at-chance information about the *distance to object*. This may suggest that both *miss* and *false alarm* trials are caused by impaired processing of information, or at least, are captured similarly by our decoding methods. The average number of *miss* trials was 58.17 (±21.63 SD) and false alarm trials was 65.94 (±21.13 SD; out of 1920 trials).

## Can we predict behavioural errors using neuroimaging?

Finally, we asked whether we could use this information to predict the behavioural outcome of each trial. To do so, we developed a new method that classified trials based on their behavioural outcomes (*correct* vs. *miss*) by asking how well a set of classifiers, pre-trained on *correct* trials, would classify the distance of the dot from the target (*Figure 6A*). To achieve this, we used a second-level classifier which labelled a trial as *correct* or *miss* based on the average accumulated accuracies obtained for that dot at every distance from the first-level decoding classifiers which were trained on *correct* trials (*Figure 6A,B*). If the accumulated accuracy for the given dot at the given distance was less than the average accuracy obtained from testing on the validation set minus a specific threshold (based on standard deviation), the testing dot (trial) was labelled as *correct*, otherwise *miss*. In this analysis, the goal was to maximise the accuracy of predicting behaviour. For that purpose, we accumulated classification accuracies along the distances. Moreover, as each classifier performs a binary classification for each testing dot at each distance, the accumulation of classification accuracies also avoided the spurious classification accuracies to drive the decision, providing smooth 'accumulated' accuracies for predicting the behaviour. As *Figure 6B* shows, there was strong evidence (BF >10)

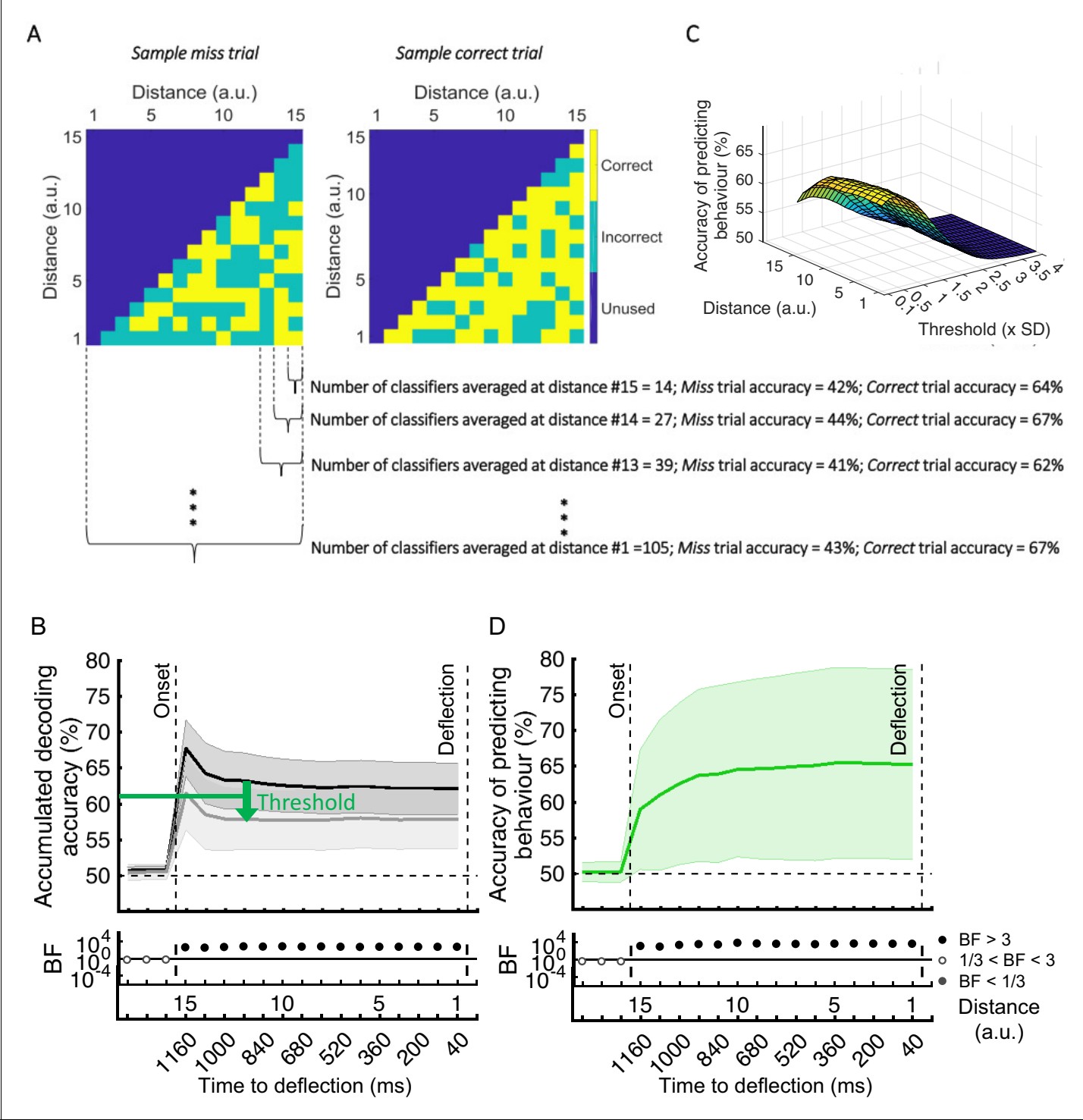

**Figure 6.** Prediction of behavioural outcome (correct vs miss) trial-by-trial using decoding of distance to object information. (**A**) Sample classifiers' accuracies (correct or incorrect classification of current distance as indicated by colours) for a miss (left panel; average accuracy ≅ 43% when the dot reached the deflection point) and a correct trial (right panel; average accuracy ≅ 67% at the deflection point). The classifiers were trained on the data from correct trials and tested on the data from correct and miss trials. For the miss trials, around half the classifiers categorised the dot's distance incorrectly by the time it reached the deflection point. (**B**) Accumulation of classifiers' accuracies over decreasing dot distances/time to deflection. This shows stronger information coding of the crucial distance to object information on the correct trials over miss trials. A variable threshold used in (**C**) is shown as a green dashed line. (**C**) Prediction of behavioural outcome as a function of threshold and distance using a second-level behavioural outcome classification. Results showed highest prediction accuracies on the participant set at around the threshold of 0.4SD under the decoding level for correct

*Figure 6 continued on next page*

*Figure 6 continued*

validation trials, increasing at closer distances. (**D**) Accuracy of predicting behavioural outcome for the left-out participant using the threshold obtained from all the other participants as function of distance/time from the deflection point. Results showed successful (~=59%) prediction of behavioural outcome of the trial as early as 80 ms after stimulus appearance. Thick lines and shading refer to average and one standard deviation around the mean across participants, respectively. Bayes factors (BF) are shown in the bottom section of each graph: Filled circles show moderate/strong evidence for either hypothesis and empty circles indicate insufficient evidence (black dots under **B** and **D**).

The online version of this article includes the following figure supplement(s) for figure 6:

**Figure supplement 1.** Accuracy of predicting behavioural outcome of trials in the early (first 5) vs late (last 5) blocks of trials before eye-blink removal.

that decoding accuracy of distances was higher for *correct* than *miss* trials with the inclusion of more classifier accuracies as the dot approached from the corner of the screen towards the centre. This clear separation of accumulated accuracies for *correct* vs. *miss* trials allowed us to predict with above-chance accuracy the behavioural outcome of the ongoing trial (*Figure 6D*). To find the optimal threshold for each participant, we evaluated the thresholds used for all other participants except for a single testing participant for whom we used the average of the best thresholds that led to highest prediction accuracy for other participants. This was ~0.4 standard deviation below the average accuracy on the other participants' validation (correct trial) sets (*Figure 6C*).

The prediction accuracy of behavioural outcome was above chance level (59% vs. 50%; BF > 10) even when the dot had only been on the screen for 80 ms, which corresponds to our furthest distance #15 (1160ms prior to deflection point; *Figure 6D*). The accuracy increased to 65.4% as the dot approached the centre of the screen, with ~64% accuracy with still 800 ms to go before required response. Importantly, the prediction algorithm showed generalisable results across participants; the threshold for decision obtained from the other participants could predict the accuracy of an independent participant's behaviour using only their neural data.

The prediction of behavioural outcome (*Figure 6*) was performed using the data from the whole data set. To test if prediction accuracy depended on the stage of the experiment, we performed the behavioural prediction procedure on data sets obtained from the first 5 (early) and the last 5 (late) stages of the experiment separately (*Figure 6—figure supplement 1*). There was no evidence for a change in the prediction power in the late vs. early blocks of trials.

## Discussion

This study developed new methods to gain insights into how attention, the frequency of target events, and the time doing a task affect the representation of information in the brain. Our new MOM task evoked reliable specific vigilance decrements in both accuracy and RT in a situation that more closely resembles real-life modern tasks than classic vigilance tasks. Using the sensitive analysis method of MVPA, we showed that neural coding was stronger for attended compared to distractor information. There was also one time-window where the interaction between the time on the task and target frequency affected decoding, with a larger decline in coding under monitoring conditions, which may reflect a neural correlate of the behavioural vigilance decrements. We also developed a novel informational brain connectivity analysis, which showed that the correlation between information coding across peri-occipital and peri-frontal areas varied with different levels of attention but did not change with errors. Finally, we utilised our recent error data analysis to predict forthcoming behavioural misses based on the neural data. In the following sections, we explain each of these findings in detail and compare them with relevant literature.

First, the MOM task includes key features of real-world monitoring situations that are not usually part of other vigilance tasks (e.g., *Mackworth, 1948*; *Temple et al., 2000*; *Beck et al., 1956*; *Rosenberg et al., 2013*), and the results show clear evidence of vigilance decrements. Behavioural performance, measured with both RT and accuracy, deteriorated over time in Monitoring (infrequent targets) relative to Active (frequent targets) conditions. One important additional advantage of the MOM task over conventional vigilance tasks is that it allows us to be specific about the vigilance decrements (by comparing Active and Monitoring conditions) separate from general time on task effects which affect *both* Active and Monitoring conditions (c.f. *Figure 3*). These vigilance decrements demonstrate that the MOM task can be used to explore vigilance in situations more closely resembling

modern environments, namely those involving moving stimuli and selection of relevant from irrelevant information, giving a useful tool for future research.

Second, the high sensitivity of MVPA to extract information from neural signals allowed us to investigate the temporal dynamics of information processing along the time course of each trial. The manipulation of attention showed a strong overall effect with enhanced representation of both the less important *direction of approach* and the most task-relevant *distance to object* information for cued dots, regardless of how frequent the targets were (*Figure 3*). The improved representation of information under attention extends previous findings from us and others (*Woolgar et al., 2015b*; *Goddard et al., 2019*; *Nastase et al., 2017*) to moving displays, in which the participants monitor multiple objects simultaneously. When targets were infrequent, modelling real-life monitoring situations, there was a strong behavioural drop in performance (i.e., vigilance effects in both accuracy and RT; *Figure 2*) and a hint in the brain activity data of a change in neural coding (namely one time-window showing evidence of an interaction between Target Frequency and Time on Task). We need more data to fully test this effect, however, our main finding is that of being able to use the difference in decoding between correct and miss trials to predict behaviour. Although the results replicated after standard eye-artefact removal, as algorithms of artefact removal are not perfect, there is still the possibility that our MEG data could be affected by some residual patterns of eye movements across conditions. In a real-world setting, it may be possible to combine information from the brain and eye-movements to further improve the prediction accuracy.

When people miss targets, they might process or encode the relevant sensory information *less* effectively than when they correctly respond to targets. This is consistent with our finding that on the majority of *miss* trials, there was *less* effective representation about the task-relevant information in the neural signal, in contrast to the consistently *more* effective representation on *correct* trials. Note that our vigilance decrement effects are defined as the difference between Active and Monitoring conditions, which allows us to be sure that we are not interpreting general task (e.g., participant fatigue) or hardware-related effects as vigilance decrements.

It is important to note that previous studies have tried other physiological/behavioural measures to determine participants' vigilance or alertness, such as pupil size (*Yoss et al., 1970*), response time variability (*Rosenberg et al., 2013*), blood pressure and thermal energy (*Lohani et al., 2019*) or even body temperature (*Molina et al., 2019*). We used highly sensitive analysis of neuroimaging data so that we could address two questions that could not be answered using these more general vigilance measures. We tested for changes in the way information is processed in the brain, particularly testing for differences in the impact of monitoring on the relevance of the information, rather than whether the participants were vigilant and alert in general. Moreover, we could also investigate how the most relevant and less relevant information was affected by the target frequency and time on the task, to find neural correlates for the behavioural vigilance decrements observed in many previous studies (e.g., *Dehais et al., 2019*; *Wolfe et al., 2005*; *Wolfe et al., 2007*; *Kamzanova et al., 2014*; *Ishibashi et al., 2012*). The less relevant information about direction of approach was modulated by attention, but its representation was not detectably affected by target frequency and time on task, and was noisier, but not noticeably attenuated, on error trials. The relative stability of these representations might reflect the large visual difference between stimuli approaching from the top left vs bottom right of the screen. In contrast, the task-relevant information of distance to object was affected by attention and was attenuated on errors. The difference might reflect the fact that only the distance information is relevant to deciding whether an item is a target, and/or the classifier having to rely on much more subtle differences to distinguish the distance categories, which collapsed over stimuli appearing on the left and right sides of the display, removing the major visual signal.

Our information-based brain connectivity method showed moderate evidence for no change in connectivity between correct and error trials. Informational connectivity is unaffected by differences in absolute levels of information encoding (e.g., lower coding on *miss* vs. *correct* trials). It could be sensitive to different levels of noise between conditions, but there was no evidence for that in this case. Apart from sensory information coding and sensory-based informational connectivity, which were evaluated here, there may be other correlates we have not addressed. Effects on response-level selection, for example, independently or in conjunction with sensory information coding, could also affect performance under vigilance conditions, and need further research.

Our connectivity method follows the recent major recent shift in literature from univariate to multivariate informational connectivity analyses (*Goddard et al., 2016*; *Karimi-Rouzbahani et al., 2017*;

*Anzellotti and Coutanche, 2018*; *Goddard et al., 2019*; *Kietzmann et al., 2019*; *Karimi-Rouzbahani et al., 2019*; *Basti et al., 2020*; *Karimi-Rouzbahani et al., 2021*). This is in contrast with the majority of neuroimaging studies using univariate connectivity analyses which can miss existing connectivity across areas when encountering low-amplitude activity on individual sensors (*Anzellotti and Coutanche, 2018*; *Basti et al., 2020*). Informational connectivity, on the other hand, is measured either through calculating the correlation between temporally resolved patterns of decoding accuracies across a pair of areas (*Coutanche and Thompson-Schill, 2013*) or the correlation between representational dissimilarity matrices (RDMs) obtained from a pair of areas (*Kietzmann et al., 2019*; *Goddard et al., 2016*; *Goddard et al., 2019*; *Karimi-Rouzbahani et al., 2019*; *Karimi-Rouzbahani et al., 2021*). Either one measures how much similarity in information coding there is between two brain areas across conditions, which is interpreted as reflecting their potential informational connectivity, and is less affected by absolute activity values compared to conventional univariate connectivity measures (*Anzellotti and Coutanche, 2018*). The method we used here evaluated the correlation between RDMs, which has provided high-dimensional information about *distance to object*, obtained from multiple sensors across the brain areas. This makes our analysis sensitive to different aspects of connectivity compared to conventional univariate analyses.

Fourth, building upon our recently developed method of error analysis (*Woolgar et al., 2019*), we were able to predict forthcoming behavioural misses based on the decoding data, before the response was given. Our method is different from the conventional method of error prediction, in which people directly discriminate correct and miss trials by feeding both types of trials to classifiers in the training phase and testing the classifiers on the left-out correct and miss trials (e.g., *Bode and Stahl, 2014*). Our method only uses correct trials for training, which makes its implementation plausible for real-world situations since we usually have plenty of correct trials and only few miss trials (i.e., cases when the railway controller diverts the trains correctly vs. misses and a collision happens). Moreover, it allows us to directly test whether the neural representations of correct trials contain information which is (on average) less observable in miss trials. We statistically compared the two types of trials and showed a reliable advantage in the level of information contained at individual-trial-level in correct vs. miss trials.

Our error prediction results showed a reliable decline in the crucial task-relevant (i.e., *distance to object*) information decoding on *miss* vs. *correct* trials but less decline in the less task-relevant information (i.e., *direction of approach*). A complementary analysis allowed the prediction of behaviourally missed trials as soon as the stimulus appeared on the screen (after ~80 ms), which was ~1160 ms before the time of response. This method was generalisable across participants, with the decision threshold for trial classification based on other participants' data successful in predicting errors for a left-out participant. A number of previous studies have shown that behavioural performance can be correlated with aspects of brain activity even before the stimulus onset (*Bode and Stahl, 2014*; *Eichele et al., 2008*; *Eichele et al., 2010*; *Weissman et al., 2006*; *Ekman et al., 2012*; *Sadaghiani et al., 2015*). Those studies have explained the behavioural errors by implicit measures such as less deactivation of the default-mode network, reduced stimulus-evoked sensory activity (*Weissman et al., 2006*; *Eichele et al., 2008*), and even the connectivity between sensory and vigilance-related/default-mode brain areas (*Sadaghiani et al., 2015*). It would be informative, however, if they could show how (if at all) the processing of task-relevant information is disrupted in the brain and how this might lead to behavioural errors. To serve an applied purpose, it would be ideal if there was a procedure to use those neural signatures to predict behavioural outcomes. Only three previous studies have approached this goal. *Bode and Stahl, 2014*, *Sadaghiani et al., 2015*, and *Dehais et al., 2019* reported maximum prediction accuracies of 62%, 63%, and 72% (with adjusted chance levels of 50%, 55%, and 59%, respectively). Here, we obtained up to 65% prediction (with a chance level of 50%), suggesting our method accesses relevant neural signatures of attention lapses, and may be sensitive in discriminating these. The successful prediction of an error from neural data more than a second in advance of the impending response provides a promising avenue for detecting lapses of attention before any consequences occur.

The overall goal of this study was to understand how neural representation of dynamic displays was affected by attention and target frequency, and whether reliable changes in behaviour over time could be predicted on the basis of neural patterns. We observed that the neural representation of critically relevant information in the brain was particularly poor on trials where participants missed the target. We used this observation to predict behavioural outcome of individual trials and showed

that we could predict behavioural outcome more than a second before action was needed. These results provide new insights about how momentary lapses in attention impact information coding in the brain and propose an avenue for predicting behavioural errors using novel neuroimaging analysis techniques.

## Materials and methods

### Participants

We tested 21 right-handed participants (10 male, 11 female, mean age = 23.4 years [SD = 4.7 years], all Macquarie University students) with normal or corrected to normal vision. The Human Research Ethics Committee of Macquarie University approved the experimental protocols and the participants gave informed consent before participating in the experiment. We reimbursed each participant AU $40 for their time completing the MEG experiment, which lasted for 2 hr including setup.

### Apparatus

We recorded neural activity using a whole-head MEG system (KIT, Kanazawa, Japan) with 160 coaxial first-order gradiometers, at a sampling rate of 1000 Hz. We projected the visual stimuli onto a mirror at a distance of 113 cm above participants' heads while they were in the MEG. An InFocus IN5108 LCD back projection system (InFocus, Portland, Oregon, USA), located outside the magnetically shielded room, presented the dynamically moving stimuli, controlled by a desktop computer (Windows 10; Core i5 CPU; 16 GB RAM; NVIDIA GeForce GTX 1060 6 GB Graphics Card) using MATLAB with Psychtoolbox 3.0 extension (*Brainard, 1997*; *Kleiner et al., 2007*). We set the refresh rate of the projector at 60 Hz and used parallel port triggers and a photodiode to mark the beginning (dot appearing on the screen) and end (dot disappearing off the screen) of each trial. We recorded participant's head shape using a pen digitiser (Polhemus Fastrack, Colchester, VT) and placed five marker coils on the head which allowed the location of the head in the MEG helmet to be monitored during the recording – we checked head location at the beginning, half way through and the end of recording. We used a fibre optic response pad (fORP, Current Designs, Philadelphia, PA, USA) to collect responses and an EyeLink 1000 MEG-compatible remote eye-tracking system (SR Research, 1000 Hz monocular sampling rate) to record eye position. We focused the eye-tracker on the right eye of the participant and calibrated the eye-tracker immediately before the start of MEG data recording.

### Task and stimuli

#### Task summary

The task was to avoid collisions of relevant moving dots with the central object by pressing the space bar if the dot passed a deflection point in a visible predicted trajectory without changing direction to avoid the central object (see *Figure 1A*; a demo can be found here https://osf.io/5aw8v/). A text cue at the start of each block indicated which colour of dot was relevant for that block. The participant only needed to respond to targets in this colour (Attended); dots in the other colour formed distractors (Unattended). Pressing the button deflected the dot in one of two possible directions (counterbalanced) to avoid collision. Participants were asked to fixate on the central object throughout the experiment.

#### Stimuli

The stimuli were moving dots in one of two colours that followed visible trajectories and covered a visual area of 3.8 × 5˚ of visual angle (dva; *Figure 1A*). We presented the stimuli in blocks of 110 s duration, with at least one dot moving on the screen at all times during the 110 s block. The trajectories directed the moving dots from two corners of the screen (top left and bottom right) straight towards a centrally presented static 'object' (a white circle of 0.25 dva) and then deflected away (either towards the top right or bottom left of the screen; in pathways orthogonal to their *direction of approach*) from the static object at a set distance (the deflection point).

Target dots deviated from the visible trajectory at the deflection point and continued moving towards the central object. The participant had to push the space bar to prevent a 'collision'. If the response was made before the dot reached the centre of the object, the dot deflected, and this was

counted as a 'hit'. If the response came after this point, the dot continued straight, and this was counted as a 'miss', even if they pressed the button before the dot totally passed through central object.

The time from dot onset in the periphery to the point of deflection was 1226 ± 10 (mean ± SD) ms. Target (and distractor event) dots took 410 ± 10 (mean ± SD) ms to cross from the deflection point to the collision point. In total, each dot moved across the display for 2005 ± 12 (mean ± SD) ms before starting to fade away after either deflection or travel through the object. The time delay between the onsets of different dots (ISI) was 1660 ± 890 (mean ± SD) ms. There were 1920 dots presented in the whole experiment (~56 min). Each 110 s block contained 64 dots, 32 (50%) in red, and 32 (50%) in green, while the central static object and trajectories were presented in white on a black background.

## Conditions

There were two target frequency conditions. In 'Monitoring' blocks, target dots were ~6.2% of cued-colour dots (2 out of 32 dots). In 'Active' blocks, target dots were 50% of cued-colour dots (16 out of 32 dots). The same proportion of dots in the non-cued colour failed to deflect; these were distractors (see *Figure 1A*, top right panel). Participants completed two practice blocks of the Active condition and then completed 30 blocks in the main experiment (15 Active followed by 15 Monitoring or *vice versa*, counterbalanced across participants).

The time between the appearance of target dots varied unpredictably, with distractors and correctly deflecting dots (events) intervening. In Monitoring blocks, there was an average time between targets of 57.88 (±36.03 SD) s. In Active blocks, there was an average time between targets of 7.20 (±6.36 SD) s.

Feedback: On target trials, if the participant pressed the space bar in time, this 'hit' was indicated by a specific tone and deflection of the target dot. There were three types of potential false alarm, all indicated by an error tone and no change in the trajectory of the dot. These were if the participant responded: (1) too early, while the dot was still on the trajectory; (2) when the dot was not a target and had been deflected automatically ('event' in *Figure 1A*, middle right); or (3) when the dot was in the non-cued colour ('distractor' in *Figure 1A*, top right) in any situation. Participants had only one chance to respond per dot; any additional responses resulted in 'error' tones. As multiple dots could be on the screen, we always associated the button press to the dot which was closest to the central object.

## Pre-processing

MEG data were filtered online using band-pass filters in the range of 0.03–200 Hz and notch-filtered at 50 Hz. We then imported the data into MATLAB and epoched them from −100 to 3000 ms relative to the trial onset time. We performed all the analyses once without and once with standard eye-artefact removal (post hoc, explained below) to see if eye movements and blinks had a significant impact on our results and interpretations. Finally, we down-sampled the data to 200 Hz for the decoding of our two key measures: *direction of approach* and *distance to object* (see below).

## Eye-related artefact removal

There are two practical reasons that the effects of eye-related artefacts (e.g. eye-blinks, saccades, etc.) should not be dominantly picked up by our classification procedure. First, the decoding analysis is time-resolved and computed in small time windows (5 ms and 80 ms, for *direction* and *distance* information decoding, respectively). For eye-related artefacts to be picked up by the classifier, they would need to occur at consistent time points across trials of the same condition, and not in the other condition, which seems implausible. Second, our MEG helmet does not have the very frontal sensors where eye-related artefacts most strongly affect neural activations (*Mognon et al., 2011*). However, to check that our results were not dominantly driven by eye-movement artefacts, we also did a post hoc analysis in which we removed these using 'runica' Independent Components Analysis (ICA) algorithm as implemented by EEGLAB. We used the ADJUST plugin (*Mognon et al., 2011*) of EEGLAB to decide which ICA components were related to eye artefacts for removal. This toolbox extracts spatiotemporal features from components to quantitatively measure if a component is related to eye movements or blinks. For all subjects except two, we identified only one component

which were attributed to eye artefacts (i.e., located frontally and surpassing the ADJUST's threshold) which we removed. For the two other participants, we identified and removed two components with these characteristics. The body of the paper presents the results of our analyses on the data without eye-artefact removal, but the corrected data can be found in the Supplementary materials.

## Multivariate pattern analyses

We measured the information contained in the multivariate (multi-sensor) patterns of MEG data by training a linear discriminant analysis (LDA) classifier using a set of training trials from two categories (e.g., for the *direction of approach* measure, this was dots approaching from left vs. right, see below). We then tested to see whether the classifier could predict the category of an independent (left-out) set of testing data from the same participant. We used a 10-fold cross-validation approach, splitting the data into training and testing subsets. Specifically, we trained the LDA classifier on 90% of the trials and tested it on the left-out 10% of the trials. This procedure was repeated 10 times each time leaving out a different 10% subset of the data for testing (i.e., 10-fold cross validation).

We decoded two major task features from the neural data: (1) the *direction of approach* (left vs. right); and (2) the distance of each moving dot from the centrally fixed object (*distance to object*), which correspond to visual (retinal) information changing over time. Our interest was in the effect of selective attention (Attended vs. Unattended) and Target Frequency conditions (Active vs. Monitoring) on the neural representation of this information, and how the representation of information changed on trials when participants missed the target.

We decoded left vs. right *directions of approach* (as indicated by yellow arrows in *Figure 1B*) every 5 ms starting from 100 ms before the appearance of the dot on the screen to 3000 ms later. Please note that as each moving dot is considered a trial, trial time windows (epochs) overlapped for 62.2% of trials. In Monitoring blocks, 1.2% of target trials overlapped (two targets were on the screen simultaneously but lagged relative to one another). In Active blocks, 17.1% of target trials overlapped.

For the decoding of *distance to object*, we split the trials into the time windows corresponding to 15 equally spaced distances of the moving dot relative to the central object (as indicated by blue lines in *Figure 1B*), with distance 1 being closest to the object, and 15 being furthest away (the dot having just appeared on the screen). Each distance covered a time window of ~80ms (varied slightly as dot trajectories varied in angle) which consisted of 4 or 5 signal samples depending on which of the 15 predetermined distances was temporally closest to each signal sample and therefore could incorporate it. Next, we concatenated the MEG signals from identical distances (splits) across both sides of the screen (left and right), so that every distance included data from dots approaching from both left and right side of the screen. This concatenation ensures that *distance* information decoding is not affected by the *direction of approach*. Finally, we trained and tested a classifier to distinguish between the MEG signals (a vector comprising data from all MEG sensors, concatenated over all time points in the relevant time window), pertaining to each pair of distances (e.g., 1 vs. 2) using a leave-one-out cross-validation procedure. As within-trial autocorrelation in signals could inflate classification accuracy (signal samples closer in time are more similar than those farther apart), we ensured that in every cross-validation run and each distance, the training and testing sets used samples from distinct sets of trials. To achieve this, trials were first allocated randomly into 10 folds, without separating their constituent signal samples. This way, the 4 or 5 signal samples from within each distance of a given trial remained together across all cross-validation runs and were never split across training and testing sets. We obtained classification accuracy for all possible pairs of distances (105 combinations of 15 distances). To obtain a single decoding value per distance, we averaged the 14 classification values that corresponded to that distance against other 14 distances. For example, the final decoding accuracy for distance 15 was an average of 15 vs. 14, 15 vs. 13, 15 vs. 12, and so on until 15 vs. 1. We repeated this procedure for our main Target Frequency conditions (Active vs. Monitoring), Attention conditions (Attended vs. Unattended), and Time on Task (first and last five blocks of each task condition, which are called early and late blocks here, respectively). This was done separately for *correct* and *miss* trials and for each participant separately.

Note that the '*direction of approach*' and '*distance to object*' information cannot be directly compared on an analogous platform as the two types of information are defined differently. There are also different number of classes in decoding for the two types of information: only two classes for

the *direction* information (left vs. right), compared to the 15 classes for the *distance* information (15 distances).

## Informational connectivity analysis

To evaluate possible modulations of brain connectivity between the attentional networks of the frontal brain and the occipital visual areas, we used a simplified version of our recently developed RSA-based informational connectivity analysis (*Goddard et al., 2016*; *Goddard et al., 2019*; *Karimi-Rouzbahani, 2018*; *Karimi-Rouzbahani et al., 2019*). Specifically, we evaluated the informational connectivity, which measures the similarity of *distance* decoding patterns between areas, across our main Target Frequency conditions (Active vs. Monitoring), Attention conditions (Attended vs. Unattended), and Time on Task (first and last five blocks of each task condition, which are called early and late blocks here, respectively). There are a few considerations in the implementation and interpretation of our connectivity analysis. First, it reflects the similarity of the way a pair of brain areas encode '*distance*' information during the whole trial. This means that we could not use the component of time in the evaluation of our connectivity as we have implemented elsewhere (*Karimi-Rouzbahani et al., 2019*; *Karimi-Rouzbahani et al., 2021*). Second, rather than a simple correlation of magnitudes of decoding accuracy between two regions of interest, our connectivity measure reflects a correlation of the *patterns* of decoding accuracies across conditions (i.e., distances here). Finally, our connectivity analysis evaluates sensory information encoding, rather than other aspects of cognitive or motor information encoding, which might have also been affected by our experimental manipulations.

Connectivity was calculated separately for *correct* and *miss* trials, using RDMs (*Kriegeskorte et al., 2008*). To construct the RDMs, we decoded all possible combinations of distances from each other yielding a 15 by 15 cross-distance classification matrix, for each condition separately. We obtained these matrices from peri-occipital and peri-frontal areas to see how the manipulation of Attention, Target Frequency, and Time on Task modulated the correlation of information (RDMs) between those areas on *correct* and *miss* trials. We quantified connectivity using Spearman's rank correlation of the matrices obtained from those areas, only including the lower triangle of the RDMs (105 decoding values). To avoid bias when comparing the connectivity on *correct* vs. *miss* trials, the number of trials were equalised by subsampling the *correct* trials to the number of *miss* trials and repeating the subsampling 100 times before finally averaging them for comparison with *miss* trials.

## Error data analysis

Next, we asked what information was coded in the brain when participants missed targets. To study information coding in the brain on *miss* trials, where the participants failed to press the button when targets failed to automatically deflect, we used our recently developed method of error data analysis (*Woolgar et al., 2019*). Essentially, this analysis asks whether the brain represents the information similarly on *correct* and *miss* trials. For that purpose, we trained a classifier using the neural data from a proportion of *correct* trials (i.e., when the target dot was detected and manually deflected punctually) and tested on both the left-out portion of the *correct* trials (i.e., cross-validation) and on the *miss* trials. If decoding accuracy is equal between the *correct* and *miss* trials, we can conclude that information coding is maintained on *miss* trials as it is on *correct* trials. However, if decoding accuracy is lower on *miss* trials than on *correct* trials, we can infer that information coding differs on *miss* trials, consistent with the change in behaviour. Since *correct* and *miss* trials were visually different after the deflection point, we only used data from before the deflection point.

For these error data analyses, the number of folds for cross-validation were determined based on the proportion of *miss* to *correct* trials (number of folds = number of *miss* trials/number of *correct* trials). This allowed us to test the trained classifiers with equal numbers of *miss* and *correct* trials to avoid bias in the comparison.

## Predicting behavioural performance from neural data

We developed a new method to predict, based on the most task-relevant information in the neural signal, whether or not a participant would press the button for a target dot in time to deflect it on a particular trial. This method includes three steps, with the third step being slightly different for the

left-out testing participant vs. the other 20 participants. First, for every participant, we trained 105 classifiers using ~80% of *correct* trials to discriminate the 15 distances. Second, we tested those classifiers using half of the left-out portion (~10%) of the *correct* trials, which we called validation trials, by simultaneously accumulating (i.e., including in averaging) the accuracies of the classifiers at each distance and further distances as the validation dot approached the central object. The validation set allowed us to determine a decision threshold for predicting the outcome of each testing trial: whether it was a *correct* or *miss* trial. Third, we performed a second-level classification on testing trials which were the other half (~10%) of the left-out portion of the *correct* trials and the *miss* trials, using each dot's accumulated accuracy calculated as in the previous step. Accordingly, if the testing dot's accumulated accuracy was **higher** than the decision threshold, it was predicted as *correct*, otherwise *miss*. For all participants, except for the left-out testing one, the decision threshold was chosen from a range of **multiples** (0.1 to 4 in steps of 0.1) of the standard deviation below the accumulated accuracy obtained for the validation set on the second step. For determining the optimal threshold for the testing participant, however, instead of a range of multiples, we used the average of the best performing multiples (i.e., the one which predicted the behavioural outcome of the trial more accurately) obtained from the other 20 participants. This avoided circularity in the analysis.

To give more detail on the second and third steps, when the validation/testing dots were at distance #15, we averaged the accuracies of the 14 classifiers trained to classify dots at distance #15 from all other distances. Accordingly, when the dot reached distance #14, we also included and averaged accuracies from classifiers which were trained to classify distance #14 from all other distances leading to 27 classifier accuracies. Therefore, by the time the dot reached distance #1, we had 105 classifier accuracies to average and predict the behavioural outcome of the trial. Every classifier's accuracies were either 1 or 0 corresponding to correct or incorrect classification of dot's distance, respectively. Note that accumulation of classifiers' accuracies, as compared to using classifier accuracy on every distance independently, provides a more robust and smoother classification measure for deciding on the label of the trials. The validation set, which was different from the testing set, allowed us to set the decision threshold based on the validation data within each subject and from the 20 participants and finally test our prediction classifiers on a separate testing set from the 21st individual participant, iteratively. The optimal threshold was 0.4 (± 0.07) times the SD below the decoding accuracy on the validation set across participants.

## Statistical analyses

To determine the evidence for the null and the alternative hypotheses, we used Bayes analyses as implemented by Krekelberg (https://klabhub.github.io/bayesFactor/) based on *Rouder et al., 2012*. We used standard rules for interpreting levels of evidence (*Lee and Wagenmakers, 2005*; *Dienes, 2014*): Bayes factors of >10 and <1/10 were interpreted as strong evidence for the alternative and null hypotheses, respectively, and >3 and <1/3 were interpreted as moderate evidence for the alternative and null hypotheses, respectively. We interpreted the Bayes factors which fell between 3 and 1/3 as reflecting insufficient evidence either way.

Specifically, for the behavioural data, we asked whether there was a difference between Active and Monitoring conditions in terms of miss rates and RTs. Accordingly, we calculated the Bayes factor as the probability of the data under alternative (i.e., difference) relative to the null (i.e., no difference) hypothesis in each block separately. In the decoding, we repeated the same procedure to evaluate the evidence for the alternative hypothesis of a difference between decoding accuracies across conditions (e.g., Active vs. Monitoring and Attended vs. Unattended) vs. the null hypothesis of no difference between them, at every time point/distance. To evaluate evidence for the alternative of above-chance decoding accuracy vs. the null hypothesis of no difference from chance, we calculated the Bayes factor between the distribution of actual accuracies obtained and a set of 1000 random accuracies obtained by randomising the class labels across the same pair of conditions (null distribution) at every time point/distance.

To evaluate the evidence for the alternative of main effects of different factors (Attention, Target Frequency, and Time on Task) in decoding, we used Bayes factor ANOVA (*Rouder et al., 2012*). This analysis evaluates the evidence for the null and alternative hypothesis as the ratio of the Bayes factor for the full model ANOVA (i.e., including all three factors of Target Frequency, Attention, and the Time on Task) relative to the restricted model (i.e., including the two other factors while excluding the factor being evaluated). For example, for evaluating the main effect of Time on Task, the

restricted model included Attention and Target Frequency factors but excluded the factor of Time on Task.

The priors for all Bayes factor analyses were determined based on Jeffrey-Zellner-Siow priors (*Jeffreys, 1961*; *Zellner and Siow, 1980*) which are from the Cauchy distribution based on the effect size that is initially calculated in the algorithm using a *t*-test (*Rouder et al., 2012*). The priors are data-driven and have been shown to be invariant with respect to linear transformations of measurement units (*Rouder et al., 2012*), which reduces the chance of being biased towards the null or alternative hypotheses.

## Acknowledgements

This work was funded by an Australian Research Council (ARC) Discovery Project grant to ANR and AW (DP170101780). AW was supported by an ARC Future Fellowship (FT170100105) and MRC intramural funding SUAG/052/G101400. H K-R was supported by Newton International Fellowship from Royal Society (NIF\R1\192608). We thank Denise Moerel, Mark Wiggins, Jeremy Wolfe, and William Helton for contributions to an earlier design of the MOM task.

## Additional information

### Funding

| Funder | Grant reference number | Author |
|---|---|---|
| Australian Research Council | DP170101780 | Anina N Rich |
| Australian Research Council | FT170100105 | Alexandra Woolgar |
| The Royal Society | NIF\R1\192608 | Hamid Karimi-Rouzbahani |
| Medical Research Council | SUAG/052/G101400 | Alexandra Woolgar |

The funders had no role in study design, data collection and interpretation, or the decision to submit the work for publication.

### Author contributions

Hamid Karimi-Rouzbahani, Conceptualization, Data curation, Formal analysis, Investigation, Visualization, Methodology, Writing - original draft, Project administration, Writing - review and editing; Alexandra Woolgar, Conceptualization, Resources, Formal analysis, Supervision, Funding acquisition, Methodology, Project administration, Writing - review and editing; Anina N Rich, Conceptualization, Formal analysis, Supervision, Funding acquisition, Methodology, Project administration, Writing - review and editing

### Author ORCIDs

Hamid Karimi-Rouzbahani (iD) https://orcid.org/0000-0003-2694-3595

### Ethics

Human subjects: The Human Research Ethics Committee of Macquarie University approved the experimental protocols and the participants gave informed consent before participating in the experiment. The approval identifier is 52020297914411.

### Decision letter and Author response

Decision letter https://doi.org/10.7554/eLife.60563.sa1
Author response https://doi.org/10.7554/eLife.60563.sa2

## Additional files

### Supplementary files

• Transparent reporting form

## Data availability

We have shared the Magnetoencephalography data (i.e. time series) as well as behavioral data in Matlab '.mat' format on the Open Science Framework website at https://osf.io/5aw8v/ with the DOI: 10.17605/OSF.IO/5AW8V. We have also uploaded a video of the "Multiple-Object-Monitoring" paradigm, developed for this study, for easier understanding of the task at the same address. The mentioned address is dedicated to this project and we will regularly update the contents to make them easier to follow for other researchers.

The following dataset was generated:

| Author(s) | Year | Dataset title | Dataset URL | Database and Identifier |
|-----------|------|---------------|-------------|-------------------------|
| Karimi-Rouzbahani H, Woolgar A, Rich A | 2021 | Neural signatures of vigilance decrements predict behavioural errors before they occur | https://osf.io/5aw8v/ | Open Science Framework, 10.17605/OSF.IO/5AW8V |

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

# Appendix 1

## Supplementary materials

Source text file for *Figure 3—figure supplement 1*

Our first analysis was to verify that our analyses could decode the important aspects of the display, relative to chance, given the overlapping moving stimuli. Here, we give the detailed results of this analysis.

We started with the information about the *direction of approach* (top left or bottom right of screen) which is a strong visual signal but not critical to the task decision. From 95 ms post-stimulus onset onwards, this visual information could be decoded from the MEG signal for all combinations of the factors: Attended and Unattended dots, both Target Frequency conditions (Active, Monitoring), and both our Time on Task durations (Early – first 5 blocks; Late – last 5 blocks; all BF > 3, different from chance).

All conditions were decodable above chance until at least 385 ms post-stimulus onset (BF > 3; *Figure 3—figure supplement 1A*), which was when the dots came closer to the centre, losing their visual difference. There was a rapid increase in information about the direction of approach between 50 ms and 150 ms post-stimulus onset, consistent with an initial forward sweep of visual information processing (*VanRullen, 2007*; *Karimi-Rouzbahani et al., 2017*; *Karimi-Rouzbahani et al., 2019*). For attended dots only (but regardless of the Target Frequency or Time on Task), the information then increased again before the deflection time, and remained different from chance until 1915 ms post-stimulus onset, which is just before the dot faded (*Figure 3—figure supplement 1A*). The second rise of decoding, which was more pronounced for the attended dots, could reflect the increasing relevance to the task as the dot approached the crucial deflection point, but it could also be due to higher visual acuity in foveal compared to peripheral areas of the visual field. The decoding peak observed after the deflection point for the attended dots was most probably caused by the large visual difference between the deflection trajectories for the dots approaching from the left vs. right side of the screen (see the deflection trajectories in *Figure 1A*).

The most task-relevant feature of the motion is the distance between the moving dot and the central object, with the deflection point of the trajectories being the key decision point. We therefore tested for decoding of *distance* information (*distance to object,* see Materials and methods). There was a brief increase in decoding of *distance to object* for attended dots across the other factors (Target Frequency and Time on Task) between the 15th and 10th distances and for the unattended dots across the other factors between 15th and the 12th distances. This corresponds to the first 400 ms for the attended dots and the first 240 ms for the unattended dots after the onset (*Figure 3—figure supplement 1B*). *Distance* decoding then dropped somewhat before ascending again as the dot approached the deflection point. The second rise of decoding, which was more pronounced for the attended dots, could reflect the increasing relevance to the task as the dot approached the crucial deflection point, but it could also be due to higher visual acuity in foveal compared to peripheral areas of the visual field. There was moderate or strong evidence that decoding of *distance* information for all attended conditions was greater than chance (50%, BF > 3) across all 15 distance levels with the exception of distance 8 in the late monitoring condition (*Figure 3—figure supplement 1B*, left panels). There were also timepoints with greater than chance decoding for the unattended conditions but these were far less consistent (*Figure 3—figure supplement 1B*, right panels).

