## [Decision Letter]

**Acceptance summary:**

In our modern work environment there are many situations where humans have to pay sustained attention in order to catch infrequent computer errors, such as while monitoring railway systems. Combining a novel multiple-object monitoring task with computationally sophisticated analyses of human magnetoencephalography (MEG) data, Karimi-Rouzbahani and colleagues find that increasing the rarity of targets leads to a worse neural representation of a crucial target feature (distance to a potential collision). They were also able to predict whether participants would catch or miss a target based on their neural data, which may prove a first step towards developing methods to pre-empt such potentially disastrous errors.

**Decision letter after peer review:**

Thank you for submitting your article "Neural signatures of vigilance decrements predict behavioural errors before they occur" for consideration by *eLife*. Your article has been reviewed by 3 peer reviewers, and the evaluation has been overseen by a Reviewing Editor and Floris de Lange as the Senior Editor. The reviewers have opted to remain anonymous.

The reviewers have discussed the reviews with one another and the Reviewing Editor has drafted this decision to help you prepare a revised submission.

As the editors have judged that your manuscript is of interest, but as described below that additional analyses are required before it is published, we would like to draw your attention to changes in our revision policy that we have made in response to COVID-19 (https://elifesciences.org/articles/57162). First, because many researchers have temporarily lost access to the labs, we will give authors as much time as they need to submit revised manuscripts. We are also offering, if you choose, to post the manuscript to bioRxiv (if it is not already there) along with this decision letter and a formal designation that the manuscript is "in revision at *eLife*". Please let us know if you would like to pursue this option. (If your work is more suitable for medRxiv, you will need to post the preprint yourself, as the mechanisms for us to do so are still in development.)

Summary:

Karimi-Rouzbahani and colleagues investigate vigilance and sustained monitoring, using a multiple-object monitoring task in combination with magnetoencephalography (MEG) recordings in humans to investigate the neural coding and decoding-based connectivity of vigilance decrements. Using computationally sophisticated multivariate analyses of the MEG data, they found that increasing the rarity of targets led to weaker decoding accuracy for the crucial feature (distance to an object), and weaker decoding was also found for misses compared to correct responses.

While the reviewers agreed the study was interesting, they also had concerns about the approach and the interpretation of the results.

Essential revisions:

1. The introduction makes it clear that the authors acknowledge that there may be multiple sources of interference contributing to declining vigilance over time: the encoding of sensory information, appropriate responses to the stimuli, or a combination of both. In the introduction, it would help if the authors review how infrequent targets affect response patterns. In addition, it would help if the theoretical approach and assumptions of the authors were explicitly stated. For instance, the a priori assumptions surrounding the connectivity analysis should be acknowledged and discussed in the interpretation of the pattern of results (e.g., p. 32, line 658). Specifically, the focus on connectivity between frontal and occipital areas seems to assume the effects are related to sensory processing alone, but this does not preclude other influences. For instance, effects could also occur on response patterns. These considerations should be added as caveats to the interpretation.

2. It is not clear what role eye fixations play here. Participants could freely scan the display, so the retinotopic representations would change depending on where the participants fixate, but at the same time the authors claim that eye position did not matter. Materials and methods, Page 11: The authors state that "We did not perform eye-blink artefact removal because it has been shown that blink artefacts are successfully ignored by multivariate classifiers as long as they are not systematically different between decoded conditions (Grootswagers et al., 2017)." This is not a sufficiently convincing argument. Firstly, the cited paper makes a theoretical argument rather than showing this empirically. Secondly, even if this were true, the frequency of eye-related artefacts seems to be of crucial importance for a paradigm that involves moving stimuli (and no fixation). There could indeed be systematic differences between conditions that are then picked up by the classifier (i.e. if more eye-blinks are related to tiredness and in turn decreased vigilance). The authors should show that their results replicate if standard artefact removal is performed on the data.

Relatedly, on page 16 the authors claim that "If the prediction from the MEG decoding was stronger than that of the eye tracking, it would mean that there was information in the neural signal over and above any artefact associated with eye movement." This statement is problematic: Firstly, such a result might only mean that prediction from MEG decoding is stronger than decoding from eye-movements, but not relate to "artefacts" in general, to which blinks would also count. Secondly, given that the signal underlying both analyses is entirely different (and the number of features), it is not valid to directly compare the results between these analyses. More detailed analyses of fixations and fixation duration on targets and distractors might indeed be strongly related to behaviour. What is decodable at a given time might just be driven by what participants are looking at.

3. One key finding was that while classifying the direction of the dots was modulated by attention, it was insensitive to many features that were captured by a classifier trained to decode the distance from the deflection. This is surprising since both are spatial features that seem hard to separate. In addition, the procedures to decode direction vs distance were very different. Do these differences still hold if the procedure used to train the two classifiers is more analogous or matched?

4. The distance classifier was trained using only correct trials. Then in the testing stage, it was generalized to either correct or miss trials. While there is a rationale for using correct trials only, could the decoding of error prediction be an artifact of the training sample, reflecting the fact that misses were not included in the training set?

5. By accumulating classifiers across time, it looks like classifier prediction improves closer to deflection. However, this could also be due to the fact that the total amount of information provided to the classifier increased. Is there a way to control for the total amount of information at different timepoints (e.g., by using a trailing window lag rather than accumulation), or contrast the classifier that derives from accumulating information with the classifier trained moment-by-moment?

6. Predicting miss trials: The implicit assumption here is that there is "less representation" for miss trials compared to correct trials (e.g., of distance to object). But even for miss trials, the representation is significantly above chance. However, maybe the lower accuracy for the miss trials resulted from on average more trials in which the target was not represented at all rather than a weaker representation across all trials. This would call into questions the interpretation of a decline in coding. In other words, on a single trial, a representation might only be present (but could result in a miss for other reasons) or not present (which would be the case for many miss trials), and the lower averages for misses would then be the result of more trials in which the information was completely absent.

It could be that the results of the subsequent analysis (predicting misses and correct responses before they occur) are in conflict with this more pessimistic interpretation. If we understand this correctly, here the classifier predicts Distance to Object for each individual trial, and Figure 6B shows that while there is a clear difference between the correct and miss trials, the latter can still be predicted above chance level but never exceed the threshold? If this is true for all single trials, this would indeed speak for a weak but "unused" representation on miss trials. But for this the authors need to show how many of the miss trials per participant had a chance-level accuracy (i.e. might be truly unrepresented), and how many were above chance but did not exceed the threshold (i.e. might have been "less represented").

7. The relationship between the vigilance decrement and error prediction. Is vigilance decrement driving the error prediction? That is, if errors increase later on, and the signal goes down, then maybe the classifier is worse. Alternatively, maybe the classifier predictions do not necessarily monotonically decrease throughout the experiment. Is the classifier equally successful at predicting errors early and late?

8. When decoding distance, active decoding declines from early to late, even though performance does not decline (or even slightly improves from early to late). This discrepancy seems hard to explain. Is this decline in classification driven by differences in the total signal from early to late?

9. Classifier performance was extremely high almost immediately after trial onset. Does the classifier perform at chance before the trial onset, or does this reflect sustained but not stimulus-specific information?

10. The connectivity analysis appears to be just a correlation of decoding results between two regions of interest. This means, if one "region" allows for decoding the distance to the object, the other one does too. However, this alone does not equal connectivity. It could simply mean that patterns across the entire brain allow for decoding the same information. For example, it would not be surprising to find that both ROIs correlate more strongly for correct trials (i.e. the brain has obviously represented the relevant information) than for errors (i.e. the brain has failed to represent the information), without this necessarily being related to connectivity at all. The more parsimonious interpretation here is that information might have been represented across all channels at this time. The authors show no evidence that only these two (arbitrarily selected) "regions" encode the information while other do not. To show evidence for meaningful connectivity, (a) the spread of information should be limited to small sub-regions, and (b) the decoding results in one "region" should predict the results in another region in time (as for DCM).

11. The display of the results is very dense, and it not always clear whether decoding for a specific variable was above chance or not. The authors often focused on relative differences, making it difficult to fully understand the meaning of the full pattern of results. The Bayes-factor plots in the decoding results figures are so cramped that it is very difficult to actually see the individual dots and to unpack all of this (e.g., Figure 3). Could this complexity be somehow reduced, maybe by dividing the panels into separate figures? The two top panels in Figure 3B should also include the chance level as in A. It looks like the accuracy is very low for unattended trials, which is only true in comparison to attended trials, but (as also shown in Supplementary Figure 1) it was clearly also encoded in unattended trials, which is very important for interpreting the results.

12. While this is methodologically interesting work, there is no convincing case made for what exactly the contribution of this study is for theories of vigilance. It seems that the findings can be reduced to that a lack of decodability of relevant target features from brain activity predicts that participants will miss the target. This alone, however, does not seem to be very novel. Even if the issues above are addressed, the study only demonstrates that with less attention to the target, there is less evidence of representations of the relevant features of targets in the brain. The authors also find the expected decrements for rare targets and when participants do not actively monitor the targets. How do these findings contribute to "theories of vigilance", as claimed by the authors?

---

## [Author Response]

Essential revisions:1. The introduction makes it clear that the authors acknowledge that there may be multiple sources of interference contributing to declining vigilance over time: the encoding of sensory information, appropriate responses to the stimuli, or a combination of both. In the introduction, it would help if the authors review how infrequent targets affect response patterns.

We added the relevant information about response patterns to the Introduction as below:

“To date, most vigilance and rare target studies have used simple displays with static stimuli. […] Overall, vigilance decrements in terms of poorer performance can be seen in both accuracy and in reaction times, depending on the task.”

In addition, it would help if the theoretical approach and assumptions of the authors were explicitly stated. For instance, the a priori assumptions surrounding the connectivity analysis should be acknowledged and discussed in the interpretation of the pattern of results (e.g., p. 32, line 658). Specifically, the focus on connectivity between frontal and occipital areas seems to assume the effects are related to sensory processing alone, but this does not preclude other influences. For instance, effects could also occur on response patterns. These considerations should be added as caveats to the interpretation.

We have now carefully reviewed the manuscript to be sure our assumptions and approach for the connectivity analyses are explicit. We have added the suggested material to the interpretation of the pattern of results, and acknowledge the potential for other influences on the connectivity results as caveats to our interpretation.

We now limit our discussion of the connectivity results as relevant to evaluating sensory aspects of information encoding (in the Materials and methods section) as below:

“There are a few considerations in the implementation and interpretation of our connectivity analysis. First, it reflects the similarity of the way a pair of brain areas encode “distance” information during the whole trial. This means that we could not use the component of time in the evaluation of our connectivity as we have implemented elsewhere (Karimi-Rouzbahani et al., 2019; Karimi-Rouzbahani et al., 2020). Second, rather than a simple correlation of magnitudes of decoding accuracy between two regions of interest, our connectivity measure reflects a correlation of the patterns of decoding accuracies across conditions (i.e., distances here). Finally, our connectivity analysis evaluates sensory information encoding, rather than other aspects of cognitive or motor information encoding, which might have also been affected by our experimental manipulations.”

We now provide the rationale and our predictions about the impact of visual and auditory attention on our connectivity metric (in the Results section) based on the literature, as below.

“In line with attentional effects on sensory perception, we predicted that connectivity between the frontal attentional and sensory networks should be lower when not attending (vs. attending; Goddard et al., 2019). Behavioural errors were also previously predicted by reduced connection between sensory and ‘vigilance-related’ frontal brain areas (Ekman et al., 2012; Sadaghiani et al., 2015). Therefore, we predicted a decline in connectivity when targets were lower in frequency, and with increased time on task, as these led to increased errors in behaviour, specifically under vigilance conditions in our task (i.e., late blocks in Monitoring vs. late blocks in Active; Figure 2).”

We have toned down our conclusions (a) and added the possibility that other factors could also contribute to our vigilance decrement effects in the Discussion, as below (b).

a) “One explanation for the decrease in decoding accuracy for task-relevant information could be that when people monitor for rare targets, they process or encode the relevant sensory information less effectively as the time passes, relative to conditions in which they are actively engaged in completing the task.”

b) “Apart from sensory information coding and sensory-based informational connectivity, which were evaluated here and provide plausible neural correlates for the vigilance decrement, there may be other correlates we have not addressed. Effects on response-level selection, for example, independently or in conjunction with sensory information coding, could also affect performance under vigilance conditions, and need further research.”

2. It is not clear what role eye fixations play here. Participants could freely scan the display, so the retinotopic representations would change depending on where the participants fixate, but at the same time the authors claim that eye position did not matter.

We did not mean to claim that eye position doesn’t matter at all, but rather that our design ensures minimal effect of eye-related artefacts on the classifiers. We have carefully revised the manuscript to ensure this is clear (detailed response and additional analyses below).

Materials and methods, Page 11: The authors state that "We did not perform eye-blink artefact removal because it has been shown that blink artefacts are successfully ignored by multivariate classifiers as long as they are not systematically different between decoded conditions (Grootswagers et al., 2017)." This is not a sufficiently convincing argument. Firstly, the cited paper makes a theoretical argument rather than showing this empirically. Secondly, even if this were true, the frequency of eye-related artefacts seems to be of crucial importance for a paradigm that involves moving stimuli (and no fixation). There could indeed be systematic differences between conditions that are then picked up by the classifier (i.e. if more eye-blinks are related to tiredness and in turn decreased vigilance). The authors should show that their results replicate if standard artefact removal is performed on the data.

We appreciate the point here. There are theoretical and practical arguments that eye-related artefacts should not drive our effects, but to be sure we also now present our results with standard artefact removal as well.

Overall increases in eye-related artefacts (such as blinks) over time-on-task would not be an issue, as our design relies on comparisons between Active and Monitoring, and so any general effects should have negligible impact. But for these comparisons, there may indeed be differences in the number of eye blinks – and in fact, these conditions involve different levels of attentional recruitment, which has previously shown to correlate with the frequency of eye blinks (Nakano et al., 2013). Thus, we certainly do not want to claim that eye-related artefacts do not matter at all, but, importantly, there are two practical reasons that the effects of eye blinks should not be dominantly picked up by our classification procedure. First, the decoding analysis is time-resolved and computed in small time windows (5 ms and 80 ms, for direction and distance information decoding, respectively). For eye blink patterns to be picked up by the classifier, they would need to occur at consistent time points across trials of the same condition, and not in the other condition, which seems implausible. Second, our MEG helmet does not have the very frontal sensors where eye-related artefacts most strongly affect neural activations (Mognon et al., 2011), but we appreciate that this does not rule out their presence altogether.

To check empirically that eye-related artefacts were not driving our effects, we re-ran our analyses with standard artefact removal as requested. We see the same pattern of results as before, for both the key task-relevant feature of distance-to-object and the less relevant feature of direction of approach. We present the full comparative analysis in Figure 3—figure supplement 2. In the paper we now state that the results replicate with artefact removal and present the additional eye-movement-corrected results in the supplementary materials.

“…,we also did a post-hoc analysis in which we removed these using “runica” Independent Components Analysis (ICA) algorithm as implemented by EEGLAB. We used the ADJUST plugin (Mognon et al., 2011) of EEGLAB to decide which ICA components were related to eye artefacts for removal. This toolbox extracts spatiotemporal features from components to quantitatively measure if a component is related to eye movements or blinks. For all subjects except two, we identified only 1 component which were attributed to eye artefacts (i.e., located frontally and surpassing the ADJUST’s threshold) which we removed. For the two other participants, we identified and removed two components with these characteristics.”

Figure 3—figure supplement 2B shows the decoding results for the key task-relevant feature of distance-to-object without and with eye-related artefact removal, in the left and right panels, respectively. The main effects of attention and time on the task and the key interaction between target frequency and time on the task remain after eye artefact removal, replicating our initial pattern of results.

Figure 3—figure supplement 2A shows the decoding results for the direction of approach information without and with eye artefact removal. The results again replicate those of the original analysis: as before there is a main effect of Attention but no main effect of Time on Task or Target Frequency, and no interaction.

We also checked to see if our trial outcome prediction (Figure 6D) could be driven by eye artefacts by repeating our prediction procedure using the eye-movement corrected MEG data. The results (Author response image 1) show that although removal of eye artefacts seems to reduce the prediction accuracy slightly overall, it only has minimal effect on the statistics, replicating our original findings. We can still predict the outcome of the trial with >80% accuracy at closer distances.

**Author response image 1. sa2fig1:** The accuracy of predicting behavioral outcome of trials without and with eye artefact removal. The results are for the left-out participant (averaged over all participants) using the threshold obtained from all the other participants as function of distance/time from the deflection point. Figure 6D shows the result without eye artefact removal and Author response image 1 with eye artefact removal. Thick lines and shading refer to average and one standard deviation around the mean across participants, respectively. Bayes Factors are shown in the bottom section of each graph: Filled circles show moderate/strong evidence for either hypothesis and empty circles indicate insufficient evidence.

In the Materials and methods section, we removed the sentence “We did not perform eye-blink artefact removal because it has been shown that blink artefacts are successfully ignored by multivariate classifiers as long as they are not systematically different between decoded conditions (Grootswagers et al., 2017).”

We also added the following explanations and the figures to the manuscript in the Results section to cover this point.

“Although eye-movements should not drive the classifiers due to our design (see Materials and methods), it is still important to verify that the results replicate when standard artefact removal is applied. We can also use eye-movement data as an additional measure, examining blinks, saccades and fixations for effects of our attention and vigilance manipulations.

First, to make sure that our neural decoding results replicate after eye-related artefact removal, we repeated our analyses on the data after eye-artefact removal (see Materials and methods), which provided analogous results to the original analysis (see the decoding results without and with artefact removal in Figure 3—figure supplement 2). Specifically, for our crucial distance to object data, the main effects of Attention and Time on Task and the key interaction between Target Frequency and Time on Task remain after eye-artefact removal, replicating our initial pattern of results.

Second, we conducted a post-hoc analysis to explore whether eye movement data showed the same patterns of vigilance decrements and therefore could explain our decoding results. We extracted the proportion of eye blinks, saccades and fixations per trial as well as the duration of those fixations from the eye-tracking data for correct trials (-100 to 1400 ms aligned to the stimulus onset time), and statistically compared them across our critical conditions (Figure 3—figure supplement 3). We saw strong evidence (BF = 4.8e8) for a difference in the number of eye blinks between attention conditions: There were more eye blinks for the Unattended (distractor) than Attended (potentially targets) colour dots. We also observed moderate evidence (BF = 3.4) for difference between the number of fixations, with more fixations in Unattended vs. Attended conditions. These suggest that there are systematic differences in the number of eye blinks and fixations due to our attentional manipulation, consistent with previous observations showing that the frequency of eye blinks can be affected by the level of attentional recruitment (Nakano et al. 2013). However, there was either insufficient evidence (0.3 < BF < 3) or moderate or strong evidence for no differences (0.1 < BF < 0.3 and BF < 0.3, respectively) between the number of eye blinks and saccades across our Active, Monitoring, Early and Late blocks, where we observed our ‘vigilance decrement’ effects in decoding. Therefore, this suggests that the main vigilance decrement effects in decoding, which were evident as an interaction between Target frequency (Active vs. Monitoring) and Time on the task (Early vs. Late) (Figure 3), were not driven by eye movements.”

Relatedly, on page 16 the authors claim that "If the prediction from the MEG decoding was stronger than that of the eye tracking, it would mean that there was information in the neural signal over and above any artefact associated with eye movement." This statement is problematic: Firstly, such a result might only mean that prediction from MEG decoding is stronger than decoding from eye-movements, but not relate to "artefacts" in general, to which blinks would also count. Secondly, given that the signal underlying both analyses is entirely different (and the number of features), it is not valid to directly compare the results between these analyses. More detailed analyses of fixations and fixation duration on targets and distractors might indeed be strongly related to behaviour. What is decodable at a given time might just be driven by what participants are looking at.

We take the point on the issues with this comparison, and so have removed the analysis from the manuscript, replacing it instead with more detailed analyses of the eye movement data:

We extracted the proportion of eye blinks, saccades and fixations per trial as well as the duration of those fixations from the eye-tracking data for correct trials (-100 to 1400 ms aligned to the stimulus onset time), and statistically compared them across our critical conditions as Figure 3—figure supplement 3. We saw strong evidence (BF=4.8e^8^) for a difference in the number of eye blinks between attention conditions: There were more eye blinks for Unattended (distractor) than Attended (potentially targets) color dots. We also observed moderate evidence (BF=3.4) for difference between the number of fixations, with more fixations in Unattended vs Attended conditions. These suggest that there are systematic differences in the number of eye blinks and fixations due to our attentional manipulation, consistent with Nakano et al., (2013). However, we observed either insufficient evidence (0.3<BF<3) or moderate to strong evidence for no difference (0.1<BF<0.3 and BF<0.3, respectively) between the number of eye blinks and saccades across our Active, Monitoring, Early and Late blocks, where we observed our ‘vigilance decrement’ effects in decoding. Consistent with the replication of the results with artefact removal presented above, this suggests that the main vigilance decrement effects in decoding, which were evident as an interaction between Target frequency (Active vs. Monitoring) and Time on the task (Early vs. Late) (Figure 3), were not driven by eye movements.

This information has also been added to the supplementary materials (Figure 3—figure supplement 3) and referred to in the manuscript (text quoted under previous bullet point).

3. One key finding was that while classifying the direction of the dots was modulated by attention, it was insensitive to many features that were captured by a classifier trained to decode the distance from the deflection. This is surprising since both are spatial features that seem hard to separate.

Yes, we see vigilance decrement effects for the distance information but not the direction of approach. Although they both rely on similar features of the visual display, the direction information classifier is likely to be driven primarily by the large visual difference between the categories (approach from the left vs approach from the right). In the key distance measure, we collapse across left and right approaching dots, which means the classifier has to use much more subtle differences (and is therefore more likely to be sensitive to other modulations). Moreover, the two types of information also differ in their importance to the task: Only the distance information is relevant to deciding whether an item is a target.

We have added to the Discussion noting this point.

“The less relevant information about direction of approach was modulated by attention, but its representation was not detectably affected by target frequency and time on task, and was noisier, but not noticeably attenuated, on error trials. The relative stability of these representations might reflect the large visual difference between stimuli approaching from the top left vs bottom right of the screen. In contrast, the task-relevant information of distance to object was affected by attention, target frequency and time on task and was dramatically attenuated on errors. The difference might reflect the fact that only the distance information is relevant to deciding whether an item is a target, and/or the classifier having to rely on much more subtle differences to distinguish the distance categories, which collapsed over stimuli appearing on the left and right sides of the display, removing the major visual signal.”

In addition, the procedures to decode direction vs distance were very different. Do these differences still hold if the procedure used to train the two classifiers is more analogous or matched?

In terms of technical differences in the decoding procedure between distance and direction information, we cannot directly compare the two types of information on an analogous platform because they have to be defined differently. There are a different number of classes in decoding for the two types of information: only two classes for the *direction* information (left vs. right), compared to the 15 classes for the distance information (15 distances). Therefore, if anything, the decoding of distance should result in less information compared to the direction of approach as the higher number of classes in decoding could potentially result in more noise in the data by decreasing signal to noise ratio per class.

We have added the following paragraph to the Materials and methods section to clarify the point:

“Note that the ‘direction of approach’ and ‘distance to object’ information cannot be directly compared on an analogous platform as the two types of information are defined differently. There are also different number of classes in decoding for the two types of information: only two classes for the direction information (left vs. right), compared to the 15 classes for the distance information (15 distances).”

4. The distance classifier was trained using only correct trials. Then in the testing stage, it was generalized to either correct or miss trials. While there is a rationale for using correct trials only, could the decoding of error prediction be an artifact of the training sample, reflecting the fact that misses were not included in the training set?

No, we do not think there is any way it could be an artefact. Our hypothesis is that correct trials contain information which is missing from miss trials. In other words, miss trials are in some way different from correct trials. Thus, it is crucial to use only correct trials in the training set. Please note that our approach is different from most conventional studies in which people directly discriminate correct and miss trials by feeding both types of trials to classifiers in the training phase and test the classifiers on the left-out correct and miss trials (i.e., without any feature extraction; as in Bode and Stahl, 2014). While this standard approach might lead to a higher classification performance, we developed our new approach for two main reasons. First, in the real world and many vigilance studies, there is usually not enough miss data to train classifiers. Second, we wanted to directly test whether the neural representations of correct trials contain some information which is (on average) less observable in miss trials. The result of conventional methods can reflect general differences between correct and miss trials (i.e., general level of attention, not time-locked to stimulus presentation), but cannot inform us about whether the difference reflects changes in information coding in the correct vs. miss trials; our approach allows this more specific inference.

In our approach, we trained our classifiers on correct trials and tested them on both correct and miss trials. Crucially, we tested the trained classifiers only on unseen data for both correct and miss trials. Specifically, when testing the classifiers, we used only the correct trials which were not used in the training phase. Therefore, there is no artefactual reason that the testing trials should be more similar to the training-phase trials for the correct compared to miss trials; the decoding prediction works because the correct testing trials have more similar neural representations to the correct training trials than the miss testing trials do.

We have added an explanation of the difference between approaches to the manuscript to ensure this point is clearer to the reader.

“Our method is different from the conventional method of error prediction, in which people directly discriminate correct and miss trials by feeding both types of trials to classifiers in the training phase and testing the classifiers on the left-out correct and miss trials (e.g., Bode and Stahl, 2014). Our method only uses correct trials for training, which makes its implementation plausible for real-world situations since we usually have plenty of correct trials and only few miss trials (i.e., cases when the railway controller diverts the trains correctly vs. misses and a collision happens). Moreover, it allows us to directly test whether the neural representations of correct trials contain information which is (on average) less observable in miss trials. We statistically compared the two types of trials and showed a large advantage in the level of information contained at individual-trial-level in correct vs. miss trials.”

5. By accumulating classifiers across time, it looks like classifier prediction improves closer to deflection. However, this could also be due to the fact that the total amount of information provided to the classifier increased. Is there a way to control for the total amount of information at different timepoints (e.g., by using a trailing window lag rather than accumulation), or contrast the classifier that derives from accumulating information with the classifier trained moment-by-moment?

Although it is likely that some of the increase in information reflects increased attention as the dot approaches the object, we think primarily that yes, the improved prediction power closer to the central object is likely to be due to accumulation of information (Figure 6D) and it will decline if we use a subsample of the accumulated information. We took this approach as the main purpose of our prediction analysis was to predict the outcome of the trial with maximal accuracy. We added the following sentence to the manuscript to clarify the point.

“In this analysis, the goal was to maximise the accuracy of predicting behaviour. For that purpose, we accumulated classification accuracies along the distances. Moreover, as each classifier performs a binary classification for each testing dot at each distance, the accumulation of classification accuracies also avoided the spurious classification accuracies to drive the decision, providing smooth “accumulated” accuracies for predicting the behaviour.”

6. Predicting miss trials: The implicit assumption here is that there is "less representation" for miss trials compared to correct trials (e.g., of distance to object). But even for miss trials, the representation is significantly above chance. However, maybe the lower accuracy for the miss trials resulted from on average more trials in which the target was not represented at all rather than a weaker representation across all trials. This would call into questions the interpretation of a decline in coding. In other words, on a single trial, a representation might only be present (but could result in a miss for other reasons) or not present (which would be the case for many miss trials), and the lower averages for misses would then be the result of more trials in which the information was completely absent.It could be that the results of the subsequent analysis (predicting misses and correct responses before they occur) are in conflict with this more pessimistic interpretation. If we understand this correctly, here the classifier predicts Distance to Object for each individual trial, and Figure 6B shows that while there is a clear difference between the correct and miss trials, the latter can still be predicted above chance level but never exceed the threshold? If this is true for all single trials, this would indeed speak for a weak but "unused" representation on miss trials. But for this the authors need to show how many of the miss trials per participant had a chance-level accuracy (i.e. might be truly unrepresented), and how many were above chance but did not exceed the threshold (i.e. might have been "less represented").

This is a really good point. Yes, in principle, the average decoding levels could be composed of ‘all or none’ misses or graded drops in information, and it is possible that on some miss trials there is a good representation but the target is missed for other reasons (e.g., a response-level error). As neural data are noisy and multivariate decoding needs cross-validation across sub samples of the data, and because each trial, at each distance, can only be classified correctly or incorrectly by a two-way classifier, we tend not to compare the decoding accuracies in a trial-by-trial manner, but rather on average (Grootswagers et al., 2017). However, if we look at an individual dataset and examine all the miss trials (averaged over the 15 distances and cross-validation runs) in our distance-to-object decoding, we can get some insights into the underlying distributions.

We show the distribution of individual trial decoding accuracies for all participants on correct (Figure 5—figure supplement 1A) and miss (Figure 5—figure supplement 1B) trials. The vertical axis shows the number of trials in each accuracy bin of the histogram and the horizontal axis shows the decoding accuracy for each trial obtained by averaging its decoding accuracies over cross-validation folds (i.e., done by subsampling the correct trials into train and test sets and repeating the procedure until all correct trials are used once as training data and once as testing data) and distances. We calculated the percentage of miss trials for which there was strong evidence (BF>10) for above-chance decoding accuracies. To do this, we generated a null distribution with 100*N trials, where we produced 1000 decoding accuracies for each trial by randomizing the labels of distances for that trial. We used the same procedure for Bayes analyses as detailed in the manuscript.

The histograms of individual miss trials suggest a single distribution centred around chance decoding or slightly above (Figure 5—figure supplement 1B). This means that on an individual miss trial, there may be higher or lower decoding, but it is nowhere near the consistent high decoding levels we see for correct trials (Figure 5—figure supplement 1A). This seems consistent with an interpretation that on (most) miss trials, information is less present than on correct trials. Presumably it is this difference that allows our second level classifier to successfully predict the behavioural outcome on >80% of trials.

In contrast, for the correct trials, all trials (100%) for all subjects showed above-chance (>50%) decoding accuracy, with average accuracies around 80%. This suggest that as opposed to missed trials, in which some trials showed some distance information and some did not, on correct trials, all trials reflect the task-related information.

In order to quantify the overlap between correct and miss trials in individual trial level (as opposed to group-level Bayes factor analysis in the manuscript (Figure 5)), we calculated the Cohen’s d (Cohen, 1969) between the two distributions. As the results show (Figure 5—figure supplement 2C), there is a large difference (d >2) between the two distributions for every participant and condition. D values were mostly higher than 3 which corresponds to less than 7% overlap between decoding accuracies obtained for the correct and miss trials.

Overall, this additional analysis demonstrates that although the miss trials vary somewhat in levels of information (as measured by decoding), with some trials representing the distance information while others do not represent the distance information at all, very few miss trials are as informative as the least informative correct trials (the distributions overlap by less than ~7%). The miss trials with high decoding are presumably those on which our second level classifier makes the wrong prediction. We have revised the description in the manuscript to make this clearer and added the following paragraph and analyses to the manuscript.

“In principle, the average decoding levels could be composed of ‘all or none’ misses or graded drops in information, and it is possible that on some miss trials there is a good representation but the target is missed for other reasons (e.g., a response-level error). As neural data are noisy and multivariate decoding needs cross-validation across sub samples of the data, and because each trial, at each distance, can only be classified correctly or incorrectly by a two-way classifier, we tend not to compare the decoding accuracies in a trial-by-trial manner, but rather on average (Grootswagers et al., 2017). However, if we look at an individual dataset and examine all the miss trials (averaged over the 15 distances and cross-validation runs) in our distance-to-object decoding, we can get some insights into the underlying distributions (Figure 5—figure supplement 1). Results showed that, for all participants, the distribution of classifier accuracies for both correct and miss trials followed approximate normal distributions. However while the distribution of decoding accuracies for correct trials was centred around 80%, the decoding accuracies for individual miss trials were centred around chance-level. We evaluated the difference in the distribution of classification accuracies between the two types of trials using Cohen’s d. Cohen’s d was approximately 3 or higher for all participants and conditions, indicating a large (d > 2; Cohen, 1969) difference between the distribution of correct and miss trials. Therefore, although the miss trials vary somewhat in levels of information, very few (< 7%) miss trials are as informative as the least informative correct trials. These results are consistent with the interpretation that there was less effective representation of the crucial information about the distance from the object preceding a behavioural miss.”

7. The relationship between the vigilance decrement and error prediction. Is vigilance decrement driving the error prediction? That is, if errors increase later on, and the signal goes down, then maybe the classifier is worse. Alternatively, maybe the classifier predictions do not necessarily monotonically decrease throughout the experiment. Is the classifier equally successful at predicting errors early and late?

Thanks for the nice question. Our error prediction results initially were obtained from the whole dataset, including all blocks of trials. To answer the reviewer’s question, we now split the blocks into the first 5 (early) and the last 5 (late) blocks and repeated the error prediction procedure on the five early and late blocks separately. To remove the potential confound of the number of trials, we equalised the number of trials across the early and late time windows. As decoding of distances decreased along the time course of the experiment on correct trials (Figure 3), we would predict that there should be less difference in decoding of correct and miss trials in the later vs earlier blocks. The new analysis bears this out: Prediction accuracy for the trial outcome (correct vs miss) declined in later stages of the experiment (moderate to strong evidence (BF>3) for higher predictability for the trial outcome in early vs. late blocks of the experiment). Importantly, even with the decline in predicting accuracy, it is still possible to predict the behavioural outcome in the late blocks with well above-chance accuracy.

We have added these results to the supplementary material of the paper (Figure 6—figure supplement 1).

“The prediction of behavioural outcome (Figure 6) was performed using the data from the whole dataset. However, it is possible that the prediction would not be as accurate in later stages of the experiment (compared to the earlier stages) as the decoding performance of the distance information declined in general in later stages (Figure 3B). To test this, we performed the behavioural prediction procedure on datasets obtained from the first 5 (early) and the last 5 (late) stages of the experiment (Figure 6—figure supplement 1). There was strong evidence for a decline in the prediction power in the late vs. early blocks of trials. However, even with the decline in prediction accuracy, it is still possible to predict the behavioural outcome in the late blocks with well above-chance accuracy (up to 75%).”

8. When decoding distance, active decoding declines from early to late, even though performance does not decline (or even slightly improves from early to late). This discrepancy seems hard to explain. Is this decline in classification driven by differences in the total signal from early to late?

Thanks for the question. We explicitly define the vigilance effects as the difference between Active and Monitoring conditions to ensure that we are not interpreting general task effects like this one as vigilance decrements. This is important because otherwise effects that are not specific to maintaining vigilance (i.e., sustaining attention in the situation where only infrequent responses are necessary) could be misinterpreted. In this case, it could be driven by a number of general factors that are not specific to vigilance such as fatigue, but also equipment effects like the MEG recording system fluctuations in baseline (e.g., due to warming up). Our crucial comparisons for both behaviour and neural correlates are the increase in ‘miss rate’ and ‘reaction time’ for Monitoring vs. Active from early to late blocks and more decline in distance decoding information (from early to late blocks) for Monitoring than for Active (Figure 3B. Interaction between Target Frequency and Time on the task). We have now added the following sentence to Discussion and amended the manuscript to ensure this is clear.

“Note that our vigilance decrement effects are defined as the difference between Active and Monitoring conditions, which allows us to be sure that we are not interpreting general task (e.g., participant fatigue) or hardware-related effects as vigilance decrements. For example, the drop in decoding over time for both Active and Monitoring that is seen in Figure 3 might reflect some of the general changes in the characteristics of the recording hardware over the course of the experiment (e.g., the MEG system warming up), but our design allows us to dissociate these from the key vigilance effects we are interested in.”

9. Classifier performance was extremely high almost immediately after trial onset. Does the classifier perform at chance before the trial onset, or does this reflect sustained but not stimulus-specific information?

Thanks for pointing out that we were missing this information – yes, the classifier performs at chance in the pre-stimulus onset time. We have now added this to the modified figures in the revised manuscript.

10. The connectivity analysis appears to be just a correlation of decoding results between two regions of interest. This means, if one "region" allows for decoding the distance to the object, the other one does too. However, this alone does not equal connectivity. It could simply mean that patterns across the entire brain allow for decoding the same information. For example, it would not be surprising to find that both ROIs correlate more strongly for correct trials (i.e. the brain has obviously represented the relevant information) than for errors (i.e. the brain has failed to represent the information), without this necessarily being related to connectivity at all. The more parsimonious interpretation here is that information might have been represented across all channels at this time. The authors show no evidence that only these two (arbitrarily selected) "regions" encode the information while other do not. To show evidence for meaningful connectivity, (a) the spread of information should be limited to small sub-regions, and (b) the decoding results in one "region" should predict the results in another region in time (as for DCM).

Thanks for the important point. Actually, our connectivity analysis is not simply a correlation of magnitudes of decoding accuracy between two regions of interest, but rather a correlation of the patterns of decoding accuracies across conditions (i.e., across distances). Our approach follows the concept of informational connectivity (explained in more detail below) which measures how much similarity in information coding there is between two brain areas across conditions, which is interpreted as reflecting their potential connectivity. Therefore, rather than the average magnitude of decoding accuracy (high vs. low), the connectivity is driven by the correlation between the patterns of decoding accuracies either across time (Coutanche and Thompson-Schill, 2013) or across conditions (Kietzmann et al., 2018). We used the latter (i.e., RDMs) here to study connectivity. This is a critical difference because high classification values in two regions will not necessarily correspond to high connectivity in our analysis.

Accordingly, the difference in classification levels between ‘correct’ and ‘miss’ trials should not determine the connectivity – it’s more the consistency of the pattern (see below example). Our connectivity relies on (Spearman’s) correlation (which normalizes absolute amplitude), and as such it is unaffected by absolute decoding values in the pairs of input vectors: connectivity will be high only if the two areas encode the information across conditions similarly rather than if they code the information very efficiently across all conditions (i.e., maximum decoding values). For example, assume that we have four brain areas A, B, C and D with (simplified and vectorized) distance RDMs (as in our work) with decoding values of [95 91 97 92], [96 98 99 94], [57 51 55 54], [58 52 59 55], respectively. The inter-area correlation/connectivity matrix would be as in Author response table 1. As you can see, a pair of brain areas with absolutely higher decoding values (A and B), but less similarity of patterns in their RDMs can led to small correlations/connectivity (0.4) while pairs of brain areas which have small decoding values but more similar patterns of decoding in their RDMs (C and D) resulted in much higher correlation/connectivity (0.8). Therefore, rather than the absolute decoding values (i.e., whether the pair of areas encode the information or not), their patterns in the RDMs determine how/if they are coding information similarly and are potentially connected.

**Author response table 1. resptable1:** Connectivity (correlation) matrix obtained from four sample areas.

AREA	*A*	*B*	*C*	*D*
*A*	1	*0.4*	0.8	1
*B*	0.4	1	0	0.4
*C*	0.8	0	1	*0.8*
*D*	1	0.4	0.8	1

Although mathematically our connectivity should be unaffected by absolute decoding values, we acknowledge that potentially noisier patterns of distance information in the brain on miss vs. correct trials could result in apparently lower connectivity for misses. We therefore added the following paragraph to the manuscript acknowledging this possibility:

“While our connectivity is unaffected by the absolute levels of information encoding in the brain on miss vs. correct trials, potentially noisier patterns of information encoding in miss (vs. correct) trials could result in the lower level of connectivity observed on miss (vs. correct) trials. Therefore, the lower level of connectivity for miss vs. correct trials observed here could result from the pair of regions representing two distinct sets of information (i.e,. becoming in some sense less connected) or representing similar information but distorted by higher level of noise.”

The more parsimonious interpretation here is that information might have been represented across all channels at this time. The authors show no evidence that only these two (arbitrarily selected) "regions" encode the information while other do not. To show evidence for meaningful connectivity, (a) the spread of information should be limited to small sub-regions, and (b) the decoding results in one "region" should predict the results in another region in time (as for DCM).

a. Yes, it is possible that the whole brain may process the information with the same pattern of decoding but changing the ROIs to smaller ones would not rule out this potential scenario (which applies to all connectivity analyses, even the conventional ones). We avoid making claims about the spatial specificity of our connectivity effect, as we are using MEG (as reflected in the names we chose for the regions: peri-occipital and peri-frontal). Please note though that these sub-regions were not arbitrary, but rather based on areas known to be involved in vision and attention, and based on previous attention work which showed a flow of information across the two areas (Goddard et al., 2016; Goddard et al., 2019).

It is very important in the interpretation of our result that, rather than making any claims about the absolute existence or magnitude of potential connectivity in the brain, we compared our connectivity indices across conditions. In other words, we do not seek to test whether connectivity exists or not between our ROIs, but rather whether any such connectivity varies with our manipulations of vigilance. Therefore, even in if the entire brain was responding similarly, the modulation of the connectivity metric is only explainable by the manipulations across our conditions.

b. We could not check the time course of our connectivity as in our previous work (Goddard et al., 2016; Karimi-Rouzbahani et al., 2020), because our distance information involves the whole trial and the direction information does not have enough number of conditions to make RDMs (please see the informational connectivity text below). Therefore, we clarified in the manuscript that:

“Informational connectivity, on the other hand, is measured either through calculating the correlation between temporally resolved patterns of decoding accuracies across a pair of areas (Coutanche and Thompson-Schill, 2013) or the correlation between representational dissimilarity matrices (RDMs) obtained from a pair of areas (Kietzman et al., 2018; Goddard et al., 2016; Goddard et al., 2019; Karimi-Rouzbahani et al., 2019; Karimi-Rouzbahani et al., 2020). Either one measures how much similarity in information coding there is between two brain areas across conditions, which is interpreted as reflecting their potential informational connectivity, and is less affected by absolute activity values compared to conventional univariate connectivity measures (Anzellotti & Coutanche, 2018).”

And added the following considerations to the methods:

“First, it reflects the similarity of the way a pair of brain areas encode “distance” information during the whole trial. This means that we could not use the component of time in the evaluation of our connectivity as we have implemented elsewhere (Karimi-Rouzbahani et al., 2019; Karimi-Rouzbahani et al., 2020). Second, rather than a simple correlation of magnitudes of decoding accuracy between two regions of interest, our connectivity measure reflects a correlation of the patterns of decoding accuracies across conditions (i.e., distances here). Finally, our connectivity analysis evaluates sensory information encoding, rather than other aspects of cognitive or motor information encoding, which might have also been affected by our experimental manipulations.”

11. The display of the results is very dense, and it not always clear whether decoding for a specific variable was above chance or not. The authors often focused on relative differences, making it difficult to fully understand the meaning of the full pattern of results. The Bayes-factor plots in the decoding results figures are so cramped that it is very difficult to actually see the individual dots and to unpack all of this (e.g., Figure 3). Could this complexity be somehow reduced, maybe by dividing the panels into separate figures? The two top panels in Figure 3B should also include the chance level as in A. It looks like the accuracy is very low for unattended trials, which is only true in comparison to attended trials, but (as also shown in Supplementary Figure 1) it was clearly also encoded in unattended trials, which is very important for interpreting the results.

We have extensively revised our figures, and expanded the Bayes plots; we hope they are now clear. We have split the panels in figures into Active and Monitoring panels, added the chance level line, and the pre-stimulus decoding values. We also reduced the density of Bayes Factor dots by down-sampling, and improved their appearance using a log scale and colour coding.

Regarding the relative differences, our design focuses on these because this allows us to be more specific about the effects that reflect actual vigilance decrements. This differs from many vigilance studies, and provides the opportunity for more specific inference. We have ensured this is clearer in the revision.

We hope the revised text and figures enhance the interpretability of the relative differences.

12. While this is methodologically interesting work, there is no convincing case made for what exactly the contribution of this study is for theories of vigilance. It seems that the findings can be reduced to that a lack of decodability of relevant target features from brain activity predicts that participants will miss the target. This alone, however, does not seem to be very novel. Even if the issues above are addressed, the study only demonstrates that with less attention to the target, there is less evidence of representations of the relevant features of targets in the brain. The authors also find the expected decrements for rare targets and when participants do not actively monitor the targets. How do these findings contribute to "theories of vigilance", as claimed by the authors?

This work makes three clear contributions to vigilance research. First, we present a novel multiple-object-monitoring paradigm that clearly evokes specific vigilance decrements in a context that mimics real-world monitoring scenarios. Our design controls for general experiment-level effects that are not specific to vigilance conditions, which as mentioned above, is surprisingly rare in the vigilance literature (which we now make clearer in the revision). This is an important contribution to the field as it provides a tool for further studies and allows us to address our hypotheses in a new and more realistic context.

Second, we showed that behavioural vigilance decrements are reflected in the neural representation of information. Previous studies have only provided coarse-grained correlates for vigilance decrements such as α-band increase in power spectrum (Kamzanova et al., 2014; Mazaheri et al., 2009; O’Connell et al., 2009). Here, we show that the neural representation of task-related information (i.e., distance) is affected by target frequency. While we agree that this is clearly a plausible prediction, it is a major step forward for a field that has had limited success in exploring specific neural correlates.

Third, we showed that change in neural representation of information between miss trials and correct trials can be used to predict the behavioural outcome on a given trial. This involves new methods that will be widely applicable, contributes to the global endeavour to link brain and behaviour, and provides a foundation for further research into potential applications for industries where detecting lapses of attention (as measured by a drop in specific task-relevant information) could prevent tragic accidents, such as rail and air traffic control.

Although we mentioned the major theories of vigilance in the paper, the theories themselves are underspecified, making it difficult to directly test them. We therefore deliberately avoided making strong claims about how our results falsified (or otherwise) the theories: they just do not contain enough specificity to do this. Nonetheless to avoid the implication that we provide a direct test of these theories, we removed the relevant paragraph in the discussion and carefully revised the paper to be explicit that the goal is not to adjudicate between the descriptive cognitive theories but rather to (a) provide a specific tool for studying vigilance in situations that mimic real-world challenges; (b) to understand what changes in the information encoded in the brain when vigilant attention lapses; and (c) to develop a method that can use neural data to predict behavioural outcomes.